# 🦩 Beyond task performance: evaluating and reducing the flaws of large multimodal models with in-context learning

**Mustafa Shukor**[1*]     **Alexandre Rame**[1]     **Corentin Dancette**[1]     **Matthieu Cord**[1,2]

[1] Sorbonne University      [2] Valeo.ai

## Abstract

Following the success of Large Language Models (LLMs), Large Multimodal Models (LMMs), such as the Flamingo model and its subsequent competitors, have started to emerge as natural steps towards generalist agents. However, interacting with recent LMMs reveals major limitations that are hardly captured by the current evaluation benchmarks. Indeed, task performances (*e.g.*, VQA accuracy) alone do not provide enough clues to understand their real capabilities, limitations, and to which extent such models are aligned to human expectations. To refine our understanding of those flaws, we deviate from the current evaluation paradigm, and (1) evaluate 10 recent open-source LMMs from 3B up to 80B parameter scale, on 5 different axes; hallucinations, abstention, compositionality, explainability and instruction following. Our evaluation on these axes reveals major flaws in LMMs. While the current go-to solution to align these models is based on training, such as instruction tuning or RLHF, we rather (2) explore the training-free in-context learning (ICL) as a solution, and study how it affects these limitations. Based on our ICL study, (3) we push ICL further and propose new multimodal ICL variants such as; Multitask-ICL, Chain-of-Hindsight-ICL, and Self-Correcting-ICL. Our findings are as follows; (1) Despite their success, LMMs have flaws that remain unsolved with scaling alone. (2) The effect of ICL on LMMs flaws is nuanced; despite its effectiveness for improved explainability, answer abstention, ICL only slightly improves instruction following, does not improve compositional abilities, and actually even amplifies hallucinations. (3) The proposed ICL variants are promising as post-hoc approaches to efficiently tackle some of those flaws. The code is available here: https://github.com/mshukor/EvALign-ICL.

## 1 Introduction

The quest for building generalist assistants has garnered significant attention and effort (OpenAI, 2023; Gao et al., 2023). The recent breakthroughs in Large Language Models (LLMs) (Brown et al., 2020; Chowdhery et al., 2022; Touvron et al., 2023b) represent a promising initial step towards this goal, achieving near-human performance across numerous NLP tasks. However, their confinement to the single textual modality remains a significant limitation in developing universal models. Consequently, the focus has shifted to building multimodal models that transcend generation and understanding across text and images (Huang et al., 2023; Yu et al., 2022; Wang et al., 2022a). The prevailing approach to develop Large Multimodal Models (LMMs), is to build on top of LLMs, bridging the gap between language and the other modalities. Those "augmented language models" (Alayrac et al., 2022; Mialon et al., 2023; Shukor et al., 2023a) beat previous models (Chen et al., 2020; Li et al., 2021; Dou et al., 2021; Shukor et al., 2022) on almost all benchmarks.

Although LMMs have achieved remarkable scores, measuring the task performance alone, such as their prediction accuracy on general benchmarks (*e.g.*, VQA accuracy or CIDEr for captioning), is insufficient to assess their genuine capabilities. For example, performances on those tasks may artificially increase simply by exploiting dataset biases and shortcuts, without truly understanding and

---

[*]Contact: {firstname.lastname}@sorbonne-unversite.fr

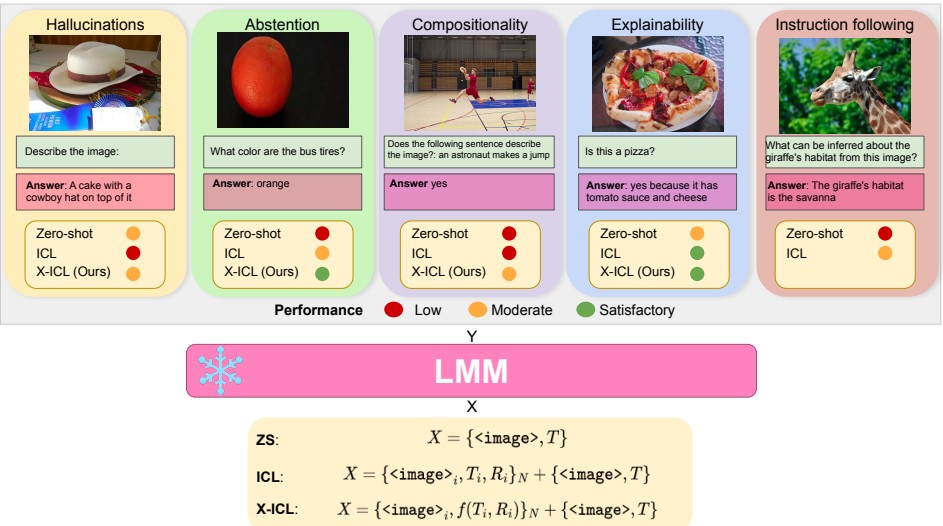

Figure 1: **Evaluation framework.** We study LMMs following 3 strategies, on different axes; hallucinations, abstention, compositionality, explainability and instruction following. In addition to an image <image> and a question T used in zero-shot (ZS), in-context learning (ICL) considers $N$ demonstrations of images-questions-answers (<image>$_i$, T$_i$, R$_i$) as input $X$, augmented by a function $f$ in our X-ICL.

generalization (Geirhos et al., 2020; Dancette et al., 2021; Du et al., 2022). While evaluating LLMs (Chang et al., 2023; Li et al., 2023d) and small multimodal models (Ma et al., 2023; Dai et al., 2023b) has received attention, the evaluation of recent LMMs has been comparatively overlooked. This is becoming increasingly important as recent works (Alayrac et al., 2022; Shukor et al., 2023b;a), in preliminary investigations, have highlighted qualitatively several major flaws (*e.g.*, hallucinations), showing that LMMs are still not aligned with the needs for deployment in real-world applications.

As argued in Askell et al. (2021), LLMs should be helpful, honest, and harmless to align with human preferences. Similarly, we argue that this should also be the case for LMMs, which becomes an urgent requirement with the exponential performance improvements. Thus, LMMs must be helpful (*e.g.*, provide explanations, follow user instructions), honest (*e.g.*, abstention or the ability to say I don't know, no hallucinations), truthful and harmless (*e.g.*, no hallucinations, especially in critical applications), generalize well and understand semantics (*e.g.*, compositionality). Thus, we start by asking the following question: *to which extent LMMs are aligned with human expectations?*

To provide an answer, we propose a different set of experiments, evaluating LMMs on 5 axes (illustrated in Figure 1). (1) Object hallucinations (OH) (honest, harmless), where LMMs generate text predictions referring to objects not present in the input image (Rohrbach et al., 2018; Dai et al., 2023b). (2) Abstention (honest), or the ability to abstain from answering, to avoid incorrect responses when the input image cannot provide the required information (Whitehead et al., 2022). (3) Compositionality (helpful, generalization) wherein the meaning of the sentence depends only on its constituents (Werning et al., 2012; Lake et al., 2017) allowing to generalize to an infinite number of compositions. Users might ask the model to (4) explain (helpful) its answers as a means to understand the underlying rationale. In addition, a true assistant should engage in conversations with users and (5) precisely follow their complex instructions (helpful) (Liu et al., 2023b). The conclusion of our study is that current LMMs lack proficiency in these aspects, revealing that scaling alone is not enough. Specifically, LMMs generate plausible and coherent answers instead of faithful and truthful ones (Section 2.1), provide answers when they do not know (Section 2.2), lack compositionality (Section 2.3), struggle to provide good explanations (Section 2.4) or precisely follow user instructions (Section 2.5).

We then investigate how to tackle these limitations. The current go-to solution to align these models is with training (*e.g.* instruction tuning, RLHF). Here, we rather focus on efficient approaches. For LLMs, a cheap, and effective alternative to finetuning is In-Context Learning (ICL), which is used to adapt the model to a particular task, a recently have been used to align LLMs (Lin et al., 2023). While ICL has been extensively investigated for LLMs (Lu et al., 2022; Liu et al., 2022; Wei et al., 2022), its application to LMMs has received less attention and mainly focuses on adaptation to new

image-text tasks (Tsimpoukelli et al., 2021; Alayrac et al., 2022). In this work, we explore to which extent we can efficiently tackle LMMs flaws using different variants of multimodal ICL. Our main contributions are:

- We evaluate 10 recent LMMs (from 3B to 80B) and show important flaws on 5 axes; object hallucinations, answer abstention, compositionality, explainability and instruction following.
- We explore Multimodal ICL as a remedy, and study its effect on these abilities. We show that while ICL can help on some aspects (explainability, abstention), it has marginal effect (instruction following), no effect (compositionality) or even worsen hallucinations.
- Based on our ICL study, we propose simple and novel ICL variants such as; Multitask-In-Context-Multitask-Learning (MT-ICL), Chain-of-Hindsight-ICL (CoH-ICL), and Self-Correcting-ICL (SC-ICL). We show the effectiveness of these variants on several abilities.

Table 1: **Evaluated LMMs**. We evaluate 10 models that differ in size, training data, and LLM initialization. Tr: training/trainable. (I): instruction. P/D: image-text pairs/web documents. ∗ use additional ChatGPT data.

| Model | # Tr. params. | # Tr. samples (P/D) | Language model | Vision Model | (I) Tuning |
|---|---|---|---|---|---|
| OFv2-3B | 1.05B | 60M/120M | MPT-1B (Team, 2023) | CLIP ViT-L/14 | ✗ |
| OFv2-3B (I) | 1.05B | 60M/120M | MPT-1B (Instruct) (Team, 2023) | CLIP ViT-L/14 | ✓ |
| OFv2-4B | 1.09B | 60M/120M | RedPajama-3B (together.ai, 2023) | CLIP ViT-L/14 | ✗ |
| OFv2-4B (I) | 1.09B | 60M∗/120M | RedPajama-3B (Instruct) (together.ai, 2023) | CLIP ViT-L/14 | ✓ |
| OFv2-9B | 1.38B | 60M∗/120M | MPT-7B (Team, 2023) | CLIP ViT-L/14 | ✗ |
| OFv1-9B | 1.31B | 5M/10M | LlaMAv1-7B (Touvron et al., 2023a) | CLIP ViT-L/14 | ✗ |
| IDEFICS-9B | 2B | 141M+/1.82B | LlaMAv1-7B (Touvron et al., 2023a) | OpenCLIP ViT-H/14 | ✗ |
| IDEFICS-9B (I) | 9B | 141M+/1.82B | LlaMAv1-7B (Touvron et al., 2023a) | OpenCLIP ViT-H/14 | ✓ |
| IDEFICS-80B | 15B | 141M+/1.82B | LlaMAv1-65B (Touvron et al., 2023a) | OpenCLIP ViT-H/14 | ✗ |
| IDEFICS-80B (I) | 80B | 141M+/1.82B | LlaMAv1-65B (Touvron et al., 2023a) | OpenCLIP ViT-H/14 | ✓ |

## 2 LMMS EVALUATION AND MULTIMODAL ICL

**Background on LMMs and ICL.** We refer by LMMs (Chen et al., 2022b; Alayrac et al., 2022; Huang et al., 2023; Li et al., 2023c) to multimodal models (beyond one modality) that train a large number of parameters (beyond 1B) on large datasets (hundreds of millions of examples). The typical development of such models builds on top of pretrained LLMs and vision encoders, with additional trainable adaptation modules. This strategy was used in the Flamingo (Alayrac et al., 2022) model, showing impressive performance on a myriad of vision-language tasks. This has driven significant efforts in the community to build similar open-source models such as Open Flamingo (OF) (Awadalla et al., 2023) and IDEFICS (Laurençon et al., 2023). The architecture of those models consists of a frozen decoder-only LLM (*e.g.*, LLaMA, MPT), frozen vision encoder (*e.g.*, CLIP-ViT) followed by a perceiver resampler, and gated cross-attention injected between LLM blocks. An interesting aspect of those LMMs is the ICL ability (Brown et al., 2020; Dong et al., 2022), allowing adaptation to new tasks with only a few demonstrations in context. Despite being heavily investigated for LLMs, as a way to solve new tasks or enhance reasoning (Wei et al., 2022; Zhang et al., 2022; Chen et al., 2022a), little work (Tsimpoukelli et al., 2021; Alayrac et al., 2022; Huang et al., 2023) addressed ICL for LMMs, which usually focus on solving general benchmarks like VQA, captioning, or classification. For multimodal ICL (M-ICL), LMMs take an input $I$ (*e.g.*, an image `<image>` and a question/instruction T), preceded by a Context $C$ (*e.g.*, $N$ task demonstrations of images and text with responses R) and generate an output $o$. M-ICL can be written as follows:

$$C = \{\langle \texttt{<image>}_i T_i R_i \texttt{<|endofchunk|>}\rangle\}_N, I = \langle \texttt{<image>}T\rangle, o = LMM([C, I]). \quad (1)$$

**Implementation details.** We consider 10 different models from OpenFlamingo (OF) (Awadalla et al., 2023) and IDEFICS (up to 80B parameters) (Laurençon et al., 2023) as described in Table 1. The models mainly change in size, initialization (LLMs), and training data. For ICL, we follow the standard way and randomly select the demonstration examples (without an explicit task instruction, results with task instructions are in Appendix I). We repeat each experiment 3 times and report the averaged results. For zero-shot, we follow other approaches and use 2 examples without images as context (*à la* Flamingo). We provide more details in Appendix C. The details for each benchmark are provided in Appendix E.

### 2.1 HALLUCINATION

Hallucinations in text is the tendency of LLMs to generate coherent plausible responses, over factual ones. By analogy, when considering multiple modalities, (Rohrbach et al., 2018) define as object

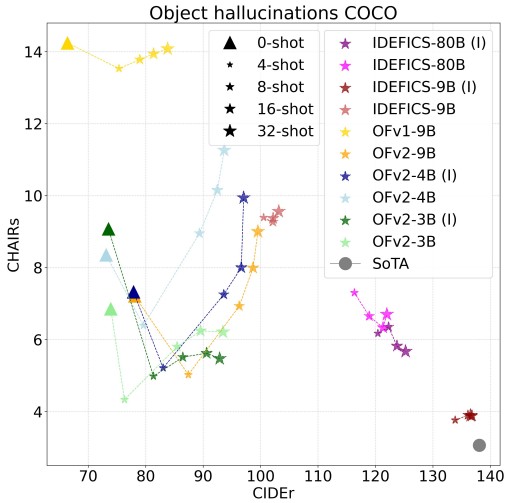
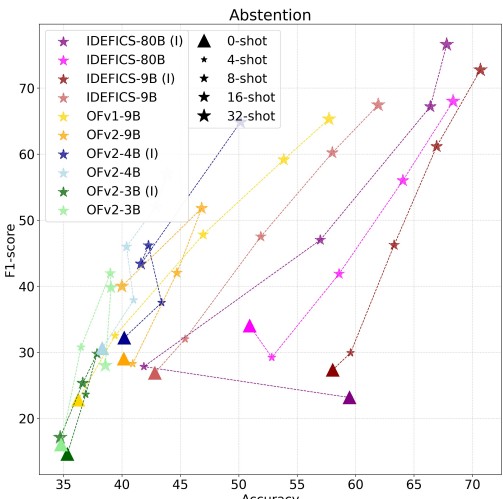

(a) **Object hallucination**. CIDEr ($\uparrow$) for captioning and CHAIR$_s$ ($\downarrow$) for hallucination on COCO dataset.

(b) **Abstention.** Overall VQA accuracy ($\uparrow$) and abstention F1-score ($\uparrow$) on TDIUC dataset.

Figure 2: Evaluation of LMMs on OH (left) and abstention (right). $\Delta$ refers to zero-shot and the $\star$ size refers to the number of shots in ICL.

hallucinations (OH) the textual description by multimodal models of objects not present in the input image. Addressing OH is critical to avoid any harm, especially in critical applications (*e.g.* autonomous driving or medical imaging).

**Benchmark.** We evaluate the LMMs on COCO captioning dataset. The performance is measured with CIDEr. In addition, to capture OH, we report the CHAIR$_s$ metric (Rohrbach et al., 2018) comparing the objects referred in the generated captioning to those actually in the image.

**LMMs suffer from object hallucinations.** Figure 2a compares the various LMMs. In zero-shot setup, all LMMs suffer from OH, as seen in the high CHAIR$_s$ scores, and in comparison to the much smaller SoTA captioning models (OFA (Wang et al., 2022b) from Shukor et al. (2023b)). This reveals that simply scaling LMMs is not enough to reduce hallucinations. For IDEFICS models, we noticed high hallucinations with zero-shot. More details and comparisons can be found in Appendix F.1.

**ICL does not reduce hallucination, but instead amplifies it.** We investigate if ICL can reduce hallucinations. We can notice (Figure 2a) that adapting models to the captioning task on COCO with 4-shots reduces OH. Yet, more than 4 shots actually amplify hallucinations, as the CHAIR$_s$ metric then increases with the number of shots. This reveals that while the overall metric (CIDEr) is improved with ICL, the generated captions contain more hallucinations. This is less the case for the largest models (IDEFICS-80B) which suffer less from such amplification.

**What reduces hallucinations?** First, pretraining on more multimodal data seems to reduce hallucinations, as all OFv2 models are better than OFv1. Second, training all model parameters (including the language model) on multimodal instruction datasets significantly reduces hallucinations (IDEFICS-9B (I) vs IDEFICS-9B). Third, instruction-tuned models (OFv1-3B (I) vs OFv1-3B and OFv1-4B (I) vs OFv1-4B) tend to hallucinate less with a higher number of ICL shots.

> *Finding* **1.** LMMs suffer from severe hallucinations. A small number of ICL shots partially alleviate it, while increasing them exacerbates the problem, especially for small models ($<$9B params.). Pretraining on more high-quality data and unfreezing the LLM weights helps to reduce hallucinations.

## 2.2 ABSTENTION

LMMs should know when they do not know, and abstain instead of providing incorrect answers. Here we study a scenario where the question can not be answered from the image.

**Benchmark.** We evaluate on TDIUC (Kafle & Kanan, 2017), a VQA dataset containing absurd questions ($\sim 22\%$ of a total number of questions), that are not related to the image and thus should not

be answered. In case of abstention, the model should generate a specific keyword ("doesnotapply"). We report the overall accuracy in addition to the F1-score abstention metric (absurd question or not).

**LMMs tend to always give an answer.** Figure 2b shows a comparison between different LMMs. From the low zero-shot F1-scores, we can notice that models are hardly able to abstain from answering to absurd questions. Adding an explicit instruction for abstention can help get additional improvements (as further shown in Appendix I).

**ICL significantly improves abstention.** Increasing the number of context examples (and thus the number of absurd examples), significantly helps abstention. However, even with the best performant model (IDEFICS-9B (I)), the F1-score is still low.

**What helps the model to abstain?** First, instruction tuning while unfreezing the language model parameters seems to significantly increase the abstention score (IDEFICS vs IDEFICS (I)). Second, increasing model size up to certain scale (9B) improves abstention (OFv2-3B vs OFv2-4B vs. OFv1-9B). In general, we notice a positive correlation between accuracy and abstention performances.

> *Finding* 2. LMMs give more likely incorrect answers than abstaining. ICL helps them abstain. Larger models, better quality data, and unfreezing LM weights improve abstention.

## 2.3 COMPOSITIONALITY

Compositionality (Hupkes et al., 2020) exists when the meaning of a sentence is determined by its elements, and the rules to compose them. To study this, we evaluate if LMMs' understanding of a caption is changed when changing its constituents.

**Benchmark Appendix E.** We evaluate on the CREPE benchmark (Ma et al., 2023) that focuses on systematicity (Fodor & Pylyshyn, 1988) and productivity (von Humboldt et al., 1988; Chomsky, 1956). It is an image-text retrieval dataset with hard negatives, constructed by changing the composition of the ground truth captions. Instead of retrieval, we create the task of Image-Text Matching (ITM) (Appendix F.2 for other choices). The model is given one caption and asked to decide whether it describes the image or not. We use the positive and negative captions provided by the benchmark. When evaluated on systematicity, we consider 2 types of negative captions: HN-Atom (replacing atoms, such as objects, attributes, or relations with atomic foils e.g., A grill underneath the porch instead of A grill on top of the porch) and HN-Comp (composing two negative captions constructed with HN-Atom e.g., A blue car and a pink toy instead of A pink car). We noticed similar observations with productivity. To complete our evaluation, we similarly evaluate on SugarCREPE (Hsieh et al., 2023) and put more details and results in Appendix F.

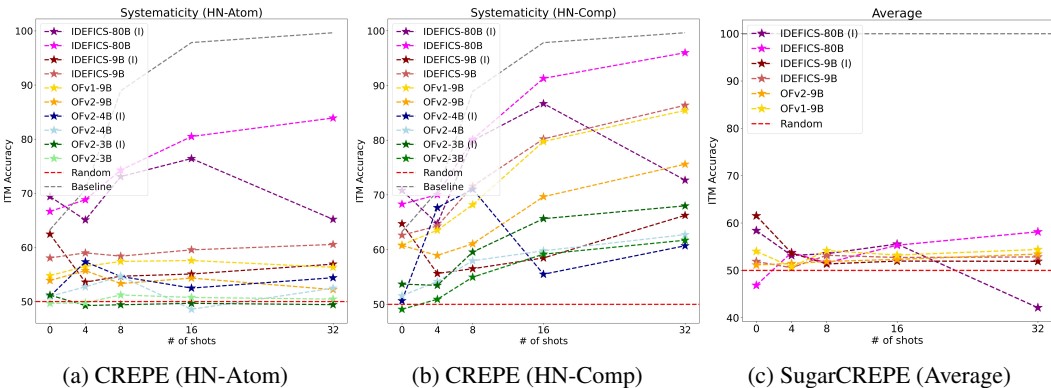

Figure 3: **Compositionality**. Models are evaluated on the CREPE and SugarCREPE with the ITM task.

**LMMs are only slightly better than random chance on compositionality.** Zero-shot performances in Figure 3 shows that LMMs are close to random on the 3 categories, with only slightly better performance on the HN-Comp. This reveals that, despite scaling the number of model parameters and of training examples, LMMs still lack compositional abilities. The baseline in Figure 3 refers to ITM without hard negative examples (Appendix F.2).

**ICL has almost no effect on atomic foils.** Interestingly, providing more demonstrations with positive and hard negative examples does not increase accuracy on the HN-Atom split. The models seem unable to detect fine-grained changes to the sentence, despite changing completely its meaning.

**ICL seems to help on compound foils.** On HN-Comp, ICL significantly increases the accuracy, especially with OFv1-9B and IDEFICS-9B (I).

**Are 80B-parameter models really good at compositionality?** In Figure 3, we can notice that the largest models (80B) seem to perform better on the CREPE benchmark. However, it is not clear if this gain is coming from really improving compositionality or exploiting biases in this benchmark, where the hard negative examples are usually longer (Ma et al., 2023), do not always make logical sense, and lack fluency Hsieh et al. (2023). Our study suggests that this improvement is coming rather from biases, which is supported in the poor performance of all LMMs on SugarCREPE Appendix F.

> *Finding* 3. LMMs lack compositional ability and struggle to acquire them even with ICL.

## 2.4 EXPLAINABILITY

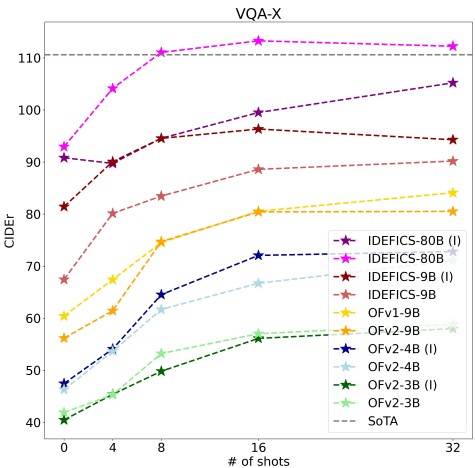

Figure 4: **Explainability**. Models are asked to generate an explanation for image, question and answer triplets from the VQA-X dataset
.

Despite the impressive abilities of LMMs, it is still unclear if generations are caused by some underlying complex reasoning based on the input image, or rather on some memorization or bias exploitation. Instead of looking at internal activations and learned features as means of output explanation, we try another and more explicit approach; by asking the model itself for an explanation.

**Benchmark.** We consider VQA-X (Park et al., 2018), a VQA dataset with human-annotated explanations for each image-question-answer triplets, and CIDEr (Vedantam et al., 2015) as the metric to measure the syntactic similarity between the generated explanations and the ground truths.

**LMMs struggle to provide good quality explanations.** To assess to which extent LMMs can explain their answers, we evaluate LMMs in a zero-shot manner. We give the model an image, a question, and the correct answer and ask it to provide a possible explanation. Figure 4 shows that LMMs can provide explanations, however, the explanation quality is very limited and significantly far from existing smaller and finetuned SoTA (Sammani et al., 2022b) (filtered scores).

**ICL significantly improves model explanations.** We evaluate the effectiveness of ICL to improve model explainability. The context consists of a few demonstrations, each one containing an image, question, correct answer, and human written explanation. Figure 4 shows that CIDEr is significantly improved by increasing the number of context demonstrations. Interestingly, while most of LMMs are still lagging, IDEFICS-80B succeed to surpass SoTA.

**Large scale models are better at explanations.** We find a clear positive correlation between model size and the quality of the generated explanation. In addition, training on more and better quality data (IDEFICS vs OF) helps to improve the performance, as well as instruction tuning with language model parameters unfrozen (IDEFICS-9B vs IDEFICS-9B (I)). However, for 80B-parameter models this is not the case, which might be due to overfitting when training the LLM.

> *Finding* **4.** LMMs still fail to provide good explanations, yet ICL can improve performances. Bigger models explain better.

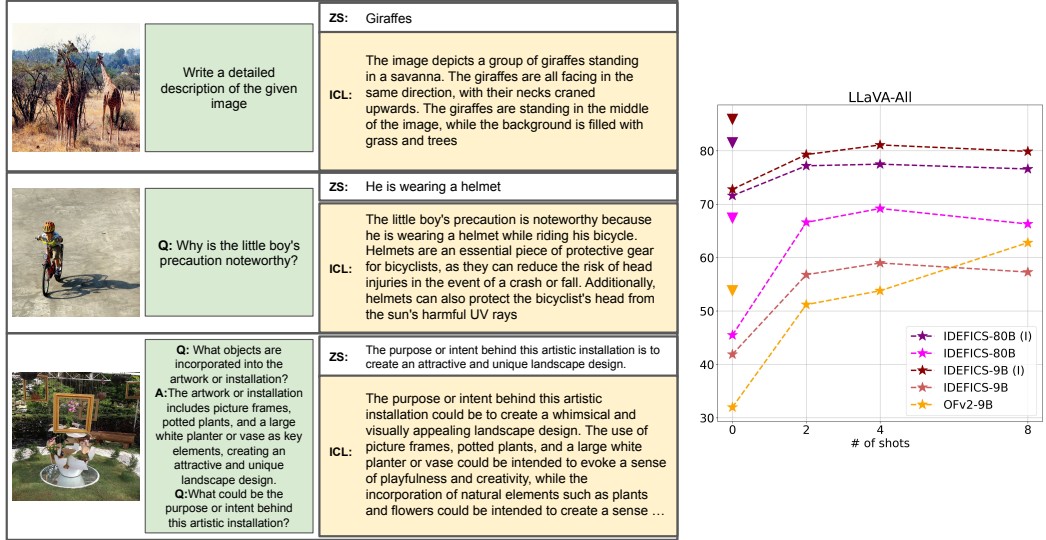

Figure 5: **Instruction following.** Evaluation on the LlaVA benchmark on 3 types of instructions: detailed descriptions, complex questions and conversations. Left: example with OFv2-9B. Right: average scores (over 3 instruction types) given by GPT-4. Detailed scores for each type in Appendix F

## 2.5 INSTRUCTION FOLLOWING

Existing multimodal models are trained to solve relatively simple tasks, such as providing shallow image descriptions or providing 1-word answers. These capabilities are not enough to build general assistants that can engage in conversation with humans. Helpful assistants should help humans answer complex questions, precisely following specific instructions and engaging in conversations. Current approaches (Liu et al., 2023b; Dai et al., 2023a) to integrate such abilities are based on instruction tuning, wherein the model is fine-tuned on curated instruction datasets. In this section, we evaluate if LMMs lack this ability and *qualitatively* investigate if ICL can help. Here we focus on IDEFICS and OFv2-9B, and provide more qualitative results in Appendix F to support our findings.

**Benchmark.** We evaluate the models on the LlaVA dataset (Liu et al., 2023b), which contains 3 types of instructions; giving detailed image descriptions, and answering complex questions and conversations. These instructions are generated with GPT-4 (text-only). For ICL, the demonstrations are selected randomly from the dataset with the same instruction type as the query. We report both qualitative and quantitative evaluation with GPT-4. (Liu et al., 2023b), GPT-4 evaluates the response and gives a score with respect to the ground truth, given also by GPT-4.

**LMMs are unable to precisely follow user instructions.** For models that are not instruction tuned, Figure 5 shows that zero-shot (ZS) LMMs lack the ability to follow user instructions. For example, short descriptions are generated even when detailed ones are explicitly asked; the simple answers do not fully answer complex questions; and the responses in the conversation are unhelpful. This is also reflected by the low ZS scores given to these models by GPT-4.

**ICL can marginally help to adapt LMMs to follow instructions.** ICL adapts the model to follow user instructions. This can be noticed in Figure 5, where the scores increase with the number of ICL shots. Qualitatively, the descriptions are more detailed; the answers to complex questions are richer and more elaborate: and the responses in conversation are more engaging. However, we also confirm here that ICL increases hallucinations, as previously shown in Section 2.1 and further discussed in Appendix J. Interestingly, we show the scores with 2-shots but without images (shown as $\nabla$), the relatively high scores raises more concerns on the effectiveness of ICL for instruction following.

> *Finding* **5.** LMMs do not precisely follow user instructions, and small number of ICL demonstrations makes them more helpful, especially for models without instruction tuning.

## 3 RECTIFYING THE FLAWS OF LMMS WITH MULTIMODAL ICL (X-ICL)

In the previous section, we show that ICL is effective in improving LMMs on some axes, such as explainability and abstention. Motivated by this, here we push ICL further and propose new improved variants to address these limitations (Appendix H for more quantitative and qualitative results).

**Chain-of-Hindsight ICL (CoH-ICL).** Chain of Hindsight (CoH) (Liu et al., 2023a) is an alternative approach for aligning LLMs to human preferences. It transforms the feedback into sentences and trains LLMs to generate this feedback. Specifically, the model is trained to generate both helpful and unhelpful responses, and during evaluation, it is prompted with the helpful prompt. Inspired by this, and to avoid costly training, we propose CoH-ICL; a training-free approach that leverages both good and bad responses as kind of in-context demonstrations. Here, we are not limited to human preferences as feedback and use positive and negative responses in general (*e.g.*, from human annotation, previous model generation, random text ...). With $T^+/R^+$ and $T^-/R^-$ referring to positive and negative demonstrations respectively, Equation (1) for CoH-ICL can be written as:

$$C = \{\langle \texttt{<image>}_i T_i T_i^+ R_i^+ T_i^- R_i^- \texttt{<|endofchunk|>} \rangle\}_N \text{ and } I = \langle \texttt{<image>} TT^+ \rangle. \quad (2)$$

Table 2: **Explainability**. Overall task accuracy and CIDEr for explanations on VQA-X. ICL here refers to single-task ICL (answer or explain).

| Model | Method | 4-shot | | 8-shot | | 16-shot | | 32-shot | |
|---|---|---|---|---|---|---|---|---|---|
| | | Acc. | CIDEr | Acc. | CIDEr | Acc. | CIDEr | Acc. | CIDEr |
| OFv2-9B | ICL | 69.52 | 61.43 | 72.71 | 74.71 | 73.11 | 80.41 | 72.93 | 80.51 |
| | CoH-ICL | – | 70.76 (+9.33) | – | 78.97 (+4.26) | – | 82.27 (+1.86) | – | 73.22 (-6.29) |
| | MT-ICL | 74.16 (+5.64) | 67.62 (+6.19) | 75.79 (+3.08) | 74.88 (+0.17) | 74.89 (+0.78) | 77.24 (-3.83) | 74.42 (+2.49) | 76.40 (-4.09) |
| IDEFICS-9B | ICL | 74.63 | 80.13 | 75.30 | 83.45 | 76.12 | 88.59 | 76.03 | 90.18 |
| | CoH-ICL | – | 82.21 (+2.08) | – | 86.85 (+3.40) | – | 89.00 (+0.41) | – | 92.18 (+2.00) |
| | MT-ICL | 74.80 (+0.17) | 81.06 (+0.93) | 76.51 (+1.21) | 83.51 (+0.06) | 76.75 (-0.63) | 83.56 (-4.56) | 78.03 (+2.0) | 85.86 (-4.32) |

*Explainability.* We leverage CoH-ICL to improve model explainability. The context consists of; an image, question, answer, human annotation as the good response, and previous model's generation (with ICL 32-shot) as the bad response. Table 2 shows significant improvements over ICL (which uses only the positive human annotations as context).

**Self-Correcting ICL (SC-ICL).** Recently, self-correction in LLMs has received large attention (Pan et al., 2023; Madaan et al., 2023; Raunak et al., 2023). The idea is to use the model itself to automatically correct its generated answers.

*Abstention.* We explore a similar approach to help LMMs abstain from answering. Specifically, we first ask the model the question using ICL. Then, for each question, we ask the model to decide whether the question is answerable based on the image or not. In case the model recognizes that the question is not answerable, the previous answer is ignored and replaced with an abstention keyword. The correction is with 32-shot in this step 2 (we consider a smaller number of shots in Appendix H.2). Following Equation (1), the steps 1 and 2 of SC-ICL can be written as: where $T^2$ is a fixed question

$$C_1 = \{\langle \texttt{<image>}_i T_i R_i \texttt{<|endofchunk|>} \rangle\}_N, I_1 = \langle \texttt{<image>} T \rangle, o_1 = LMM([C_1, I_1]), \quad (3)$$
$$C_2 = \{\langle \texttt{<image>}_i T^{2"} T_i" R^2 \texttt{<|endofchunk|>} \rangle\}_N, I_2 = \langle \texttt{<image>} T^{2"} T \rangle, o_2 = LMM([C_2, I_2]),$$

to ask the model if the following question $T_i$ is relevant to the image, and $R^2$ is yes or no. The final answer is given as a function $F$ of $o_1$ and $o_2$, i.e., $o = F(o_1, o_2)$. Table 3 shows the results with SC-ICL (32shot). We notice that SC-ICL improves significantly over ICL for both models.

Table 3: **Abstention**. Abstaining from answering unanswerable questions. We report the overall accuracy (Acc), and abstention F1-score (Abs F1) on the TDIUC dataset.

| Model | Method | 4-shot | | 8-shot | | 16-shot | | 32-shot | |
|---|---|---|---|---|---|---|---|---|---|
| | | Acc. | Abst F1 | Acc. | Abst F1 | Acc. | Abst F1 | Acc. | Abst F1 |
| OFv2-9B | ICL | 40.93 | 28.27 | 44.71 | 42.02 | 46.83 | 51.80 | 46.63 | 56.44 |
| | SC-ICL (32shot) | 44.38 (+3.45) | 47.34 (+19.07) | 46.92 (+2.21) | 52.85 (+10.83) | 48.38 (+1.35) | 57.41 (+4.67) | 47.86 (+5.61) | 59.93 (-1.64) |
| | MT-ICL | 47.99 (+7.06) | 29.99 (+1.72) | 48.41 (+3.70) | 48.09 (+6.07) | 49.13 (+2.30) | 54.58 (+2.78) | 48.83 (+2.20) | 59.14 (+2.70) |
| IDEFICS-9B | ICL | 45.41 | 32.00 | 51.89 | 47.51 | 58.01 | 60.22 | 61.94 | 67.45 |
| | SC-ICL (32shot) | 49.56 (+4.15) | 49.56 (+17.56) | 54.75 (+2.86) | 57.76 (+10.25) | 59.21 (+1.20) | 64.16 (+3.94) | 62.77 (+0.83) | 68.96 (+1.51) |
| | MT-ICL | 48.30 (+2.89) | 37.82 (+5.82) | 51.80 (-0.09) | 48.69 (+1.18) | 54.76 (-3.25) | 59.55 (-0.67) | 58.51 (-3.43) | 67.57 (+0.12) |

**Multitask ICL (MT-ICL).** Multitask learning (Caruana, 1997) aims at leveraging the synergy between tasks, usually by training one model on different related tasks. Different from this, we propose to do multitask learning in context, without changing the model's weights. Our objective is to benefit from information from other tasks to reduce LMMs flaws. With $T_i^j R_i^j$ referring to task $j$, the context $C$ in Equation (1) for MT-ICL can be written as:

$$C = \{\langle \texttt{<image>}_i \mathrm{T}_i^1 \mathrm{R}_i^1 \mathrm{T}_i^2 \mathrm{R}_i^2 \texttt{<|endofchunk|>}\rangle\}_N \text{ and } I = \langle \texttt{<image>} \mathrm{T}^1\rangle. \tag{4}$$

For *explainability*, we ask the model to simultaneously; answer the question and explain its answers preceded with the prompt "because" (we find it better to provide the answer first). With MT-ICL (Table 2) both VQA accuracy and CIDEr are better than single task (ICL). However, we notice some degradation in CIDEr with a higher number of shots. For abstention, the main task is to answer the question and the second auxiliary task is to decide whether the question is relevant to the image. Table 3 shows a significant improvement compared to single task ICL (only answering the question).

## 4 RELATED WORK

**Limitations of multimodal models.** Efforts have been made to address object hallucinations (Rohrbach et al., 2018) by designing better training objectives (Dai et al., 2023b), incorporating object labels as input (Biten et al., 2022) or costly multi-turn reasoning (Xu et al., 2023). To abstain from answering, recent work has attempted to tackle this problem by training selection functions on top of a VQA model (Whitehead et al., 2022; Dancette et al., 2023). The challenge of compositionality has received significant attention, and multiple evaluation benchmarks have been proposed (Ma et al., 2023; Thrush et al., 2022; Zhao et al., 2022). Some solutions involve training on hard negative examples (Yuksekgonul et al., 2022) or employing improved architectures (Ray et al., 2023). The issue of explainability has been tackled in various ways, such as training auxiliary models to provide explanations (Kayser et al., 2021; Marasović et al., 2020; Wu & Mooney, 2019), or training models that generate both answers and explanations (Sammani et al., 2022a). Furthermore, multimodal models also struggle to follow complex user instructions, as shown in recent work (Liu et al., 2023b; Shukor et al., 2023b). To address this, previous work fine-tune models on instruction tuning datasets (Liu et al., 2023b; Xu et al., 2022; Dai et al., 2023a; Li et al., 2023a; Zhu et al., 2023a). However, current approaches to address these limitations are focused mostly on small specialized multimodal models, and based on expensive finetuning; our ICL solutions are easier and cheaper.

**Evaluation of LMMs.** To achieve a more nuanced evaluation of different model abilities, concurrent works have proposed several benchmarks (Xu et al., 2023; Li et al., 2023b; Yu et al., 2023; Liu et al., 2023c; Yin et al., 2023). These works span evaluating multimodal models on modality comprehension (Li et al., 2023b), different capabilities (Xu et al., 2023) fine-grained tasks (Liu et al., 2023c), complicated tasks (Yu et al., 2023) or high-level 3D tasks (Yin et al., 2023). However, these benchmarks remain focused on task performance, with novelty in creating more fine-grained tasks. Besides, we differ from these benchmarks, as we consider different LMMs with ICL ability, and focus more on limitations/alignment in the context of ICL. In general, there is still a notable lack of work evaluating the limitations of LMMs.

## 5 DISCUSSION

**Reproducibility statement.** Each experiment is repeated 3 times with different context demonstrations. We use public datasets and official open-source implementations provided by respective authors. We release the code and detailed technical instructions to reproduce the results (Appendix D).

**Limitations.** The work has some limitations, further discussed in Appendix J and Appendix A, such as the limited range of abilities that we evaluate and the limited effectiveness of ICL as a partial solution for the studied flaws and models.

**Conclusion.** We evaluate the limitations of recent LMMs on different axes; object hallucination, answer abstention, compositionality, explainability and instruction following. Despite their scale, we find that LMMs still struggle on most of these axes. Besides, we study how ICL can affect these limitations, and find that while it might help on some abilities (*e.g.*, abstention and explainability and instruction following) it can amplify the flaws of LMMs (*e.g.*, hallucination) or has almost no effect at all (*e.g.*, compositionality). We also propose simple ICL variants that help reducing some of the flaws. Yet, we find that the improvements coming from ICL are limited, and more complex ICL variants or other strategies, such as RLHF might be required. Finally, we hope this provides more insights about the limitations of current LMMs, and offer promising directions towards efficiently aligning foundation models (Lin et al., 2023; Li et al., 2023e) to human preferences and expectations.

**Acknowledgments** This work was partly supported by ANR grant VISA DEEP (ANR-20-CHIA-0022), and HPC resources of IDRIS under the allocation 2022-[AD011013415] and 2023-[AD011013415R1] made by GENCI. We thank Hugo Laurencon for fruitful discussions.

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

---

## Supplementary material

---

This supplementary material is organized as follows:

## A  DISCUSSION

**Other limitations and evaluation axes.**   The work does not consider all existing limitations. For instance, other kinds of hallucinations, beyond objects (*e.g.*, relations, actions, attributes). For answer abstention, we consider the case when the question is not relevant to the image, but not for example when the question is relevant but unanswerable, or when it requires external knowledge that the model does not know. Other important axes include evaluating the reasoning ability of these models, especially in real situations (*e.g.*, embodiment) and to which extent the model prediction is grounded in the real world.

**ICL as a way to address foundation model limitations.**   Despite being effective in some benchmarks, ICL is still limited in addressing some flaws. The different variants that we propose bring additional improvements. However, more effort should be put into devising more effective variants to obtain reasonable performance. In addition, we noticed that the design of the prompt affects the results, thus more prompt engineering work can help to get additional improvement. The importance of such training-free, post-hoc approaches is, in addition to being efficient, they can be complementary to other training-based ones, such RLHF (Christiano et al., 2017; Bai et al., 2022a) and RLAIF (Bai et al., 2022b). Finally, more effort should be put into understanding why and when ICL works, to help develop better approaches.

**Other LMMs and foundation models.**   The work addresses one kind of LMMs that are based on the Flamingo architecture. We choose these models, as they obtain the best performance on several multimodal benchmarks, they are open source and exist with different scales. The work can straightforwardly be extended to other multimodal models that have ICL abilities. For the broader family of multimodal models, especially the instruction-tuned ones, we believe that these models are also flawed, and it is important to quantitatively assess their limitations. Besides LMMs, the proposed ICL variants might be also effective in tackling the limitations of LLMs, which have received great attention in recent years.

**Beyond 9B parameters.**   In this work, we only consider models up to 9B parameters. The effectiveness of ICL is limited on some benchmarks probably due to the model size. In fact, the ICL performance of OF models is not very stable as shown in the original paper (Awadalla et al., 2023) (*e.g.*, sometimes increasing the number of shots decreases the performance on VQA). Thus, it will be interesting to evaluate larger and more powerful models. In addition, as ICL becomes more effective with larger models, X-ICL approaches must be also the case, especially on benchmarks where we noticed positive correlations between scaling and performance. On harder problems such as compositionality, or hallucinations it is uncertain if ICL will become more effective.

**Beyond image-text modalities.** While this work addresses image-text models, we argue that similar limitations also exist in models trained on other modalities. We believe the extension of this work, especially the ICL part, is straightforward to models tackling other modalities (*e.g.*, videos-text or audio-text) and have ICL abilities. In fact, we argue that most of the findings on image-text models also hold on other modalities, which is supported by recent works (Shukor et al., 2023a; Girdhar et al., 2023; Shukor et al., 2023b; Zhang et al., 2023) demonstrating the feasibility of extending image-text models or using almost the same image-text techniques to address other modalities.

**Performance saturation after large number of ICL demonstrations.** In our study, we notice that the performance start to saturate after large number of shots (16/32) on most of the benchmarks. This issue can be seen in several previous work, in particular, the original work of OpenFlamingo (Awadalla et al., 2023) and IDEFICS (Laurençon et al., 2023). For example, in (Awadalla et al., 2023); the VQA accuracy saturates or even degrades after 4/8 shots. Similarly for IDEFICS, but slightly better. There is multiple possible reasons for why multimodal ICL is not as effective as in LLMs, such as: (a) the multimodal datasets are still an order of magnitude smaller than those for LLMs. In addition, the web documents used to train such models do not contain many interleaved image-text pairs (a lot less than 32), which might hinder the ability of the model to generalize to larger number of in-context demonstrations during test. b) The trainable parameters during pretraining, are relatively small (¡15B), and acquiring better ICL ability might require training more parameters for more iterations. Finally, we would like to highlight the lack of in depth analysis of ICL in the context of LMMs, which we keep for future work.

## B    RELATED WORK

**LMMs.** The success of Large Language Models (LLMs) (Brown et al., 2020; Chowdhery et al., 2022; Hoffmann et al., 2022; Touvron et al., 2023b) has spurred considerable efforts to extend the potential of these models to more modalities (Chen et al., 2022b; 2023; Huang et al., 2023; Li et al., 2023c; Wang et al., 2022c). In particular, Large Multimodal Models (LMMs) (Alayrac et al., 2022), or multimodal models (beyond one modality) that train a large number of parameters (beyond 1B parameter) on large datasets (hundreds of millions of examples). Typical LMMs build on top of LLMs, with additional adaptation modules. These models mainly differ in the adaptation modules (Shukor et al., 2023a; Li et al., 2023c), pretraining data (Schuhmann et al., 2021; Zhu et al., 2023b; Laurençon et al., 2023), and initialization (LLMs). These LMMs surpass the performance of traditional finetuned multimodal models (Li et al., 2021; Shukor et al., 2022; Dou et al., 2021). Recently, a proprietary model called Flamingo (Alayrac et al., 2022), has been proposed, followed by several open source models such as Open Flamingo (OF) (Awadalla et al., 2023) and IDEFICS (Laurençon et al., 2023). While most LMMs are currently tailored to image-text tasks, many works have demonstrated the potential for extension to other modalities (Shukor et al., 2023a; Girdhar et al., 2023; Shukor et al., 2023b; Zhang et al., 2023).

**ICL.** One of the emerging abilities when scaling LLMs, is In Context Learning (ICL) (Brown et al., 2020; Dong et al., 2022); the ability to adapt the model from demonstrations. Several works target the design of the context prompt to enhance ICL effectiveness (Lu et al., 2022; Liu et al., 2022; Zhao et al., 2021), and improve the model's reasoning ability (Wei et al., 2022; Zhang et al., 2022; Chen et al., 2022a). Few works have used ICL for aligning LLMs with human preferences, such as generating safer dialogue (Meade et al., 2023) and producing harmless, honest, and helpful text (Askell et al., 2021). However, the investigation of ICL in the realm LMMs remains limited, where previous studies (Tsimpoukelli et al., 2021; Alayrac et al., 2022; Huang et al., 2023) mainly focused on adapting pretrained LMMs to solve general benchmarks like VQA, captioning, or classification.

## C    BACKGROUND ON LMMS AND BASELINE MODELS

We consider 10 different LMMs from OpenFlamingo (OF) (Awadalla et al., 2023) and IDEFICS (Laurençon et al., 2023) as described in Table 1. For OF models; the multimodal pretraining of all models are done on part of the documents from the Multimodal-C4 dataset (Zhu et al., 2023b) and image-text pairs from the english LAION 2B (Schuhmann et al., 2022). OFv2-4B models are trained additionally on ChatGPT-generated data. Note that, the first version of OF (OFv1-9B) is trained

on less data compared to OFv2 models. For IDEFICS; the multimodal pretraining is done on data from OBELICS (Laurençon et al., 2023), LAION (Schuhmann et al., 2022), Wikipedia (Foundation) and PMD (Singh et al., 2022). IDEFICS (I) is trained additionally on several instruction-tuning datasets. The architectures of all models are similar, with the main difference in the model size and initialization (which LLM). Specifically, these models consist of a frozen decoder-only LLM (*e.g.*, LLaMA, MPT), frozen vision encoder followed by a perceiver resampler (*e.g.*, CLIP-ViT) and gated cross-attention injected between LLM blocks. The learnable gate in cross-attentions helps to stabilize the early stage of the training.

## D    EVALUATION SETUP

The evaluation of all models are done with zero-shot (a la Flamingo; 2-shot without images) or few-shot ICL, without any finetuning. In the paper, when we refer to evaluation we usually mean to the zero-shot setup. For efficient inference, we use the accelerate library (Gugger et al., 2022) from transformers, and run all OF models with float16 (which leads to very small degradation in performance compared to running with float32). For IDEFICS the inference is done with Bfloat16. For ICL, we follow the standard approach and randomly select the examples from the corresponding datasets. For each benchmark, we randomly sample a subset of examples and divide them into separate query and context examples. Each score that we report is the average of scores after repeating the experiment 3 times. We use the official open-source implementation provided by the models' authors.

## E    BENCHMARKS AND METRICS

**COCO (Lin et al., 2014) (object hallucination)**    is a widely used image captioning dataset. It consists of 118K images for training and 5K for validation and testing. Each image is human-annotated with 5 different captions. We use 5K examples from the validation set. This dataset is used to evaluate object hallucinations with the CHAIR metrics (Rohrbach et al., 2018). These metrics are based on comparing the textual objects in the generated captions to the actual objects present in the image (from the segmentation annotation of COCO images).

**TDIUC (Kafle & Kanan, 2017) (abstention)**    is a VQA dataset with 168K images and 1.6M questions divided into 12 types. The questions are imported from COCO, VQA, and Visual Genome in addition to some annotated questions. One type of them is absurd questions (366K nonsensical queries about the image). We sample 8K examples ( 22% of them absurd questions) for evaluation. To report the abstention metrics, we use the same metrics used in binary classification; accuracy and F1-score which is the harmonic mean of the precision and recall.

**CREPE (Ma et al., 2023) (compositionality)**    is a large-scale benchmark to evaluate compositionality (productivity and systematicity) in vision-language models. Based on the visual genome dataset, they propose an automated pipeline to generate hard negative captions. In this work, we focus on systematicity. For HN-Atom, the hard negatives are created by replacing the objects, attributes, and relationships in the ground truth captions with an atomic foil (*e.g.*, antonyms). For HN-Comp, they concatenate two compounds, and each one of them contains an atomic foil. We evaluate on 5K examples, randomly sampled from a test set designed for LAION (as the evaluated models use LAION during pretraining). The main difference to our work is that instead of image-text retrieval, we consider this benchmark as image-text matching (ITM) or image-text selection (ITS; where the model is given a correct and incorrect caption and the task is to select which one describes the image). For these created tasks, we report the binary classification accuracy (*e.g.*, for ITM if the caption describes the image or not). We stick to the accuracy as we sample balanced context demonstrations.

**SugarCREPE (Hsieh et al., 2023).**    Is a benchmark to remedy the previous hackable datasets, by reducing the biases and shortcuts that can be exploited when evaluating compositionality. This is mainly due to using LLMs instead of rule-based templates to create hard negative examples. It covers 7 types of hard negatives; replace (objects, attributes, and relations), swap (objects and attributes) and add (objects and attributes). Each image is associated with a positive description (image caption) and several hard negative descriptions.

**VQA-X (Park et al., 2018) (explainability)** is based on the VQA and VQAv2 datasets, and contains 32K question/answer pairs and 41K explanations annotated by humans. The explanations are intended to explain the ground truth answer for the question, based on the corresponding image. We use the test set of this benchmark (1.9K pairs and 5.9K explanations). To evaluate the explainability performance, we consider captioning metrics such as CIDEr that are based on the syntactic similarity between the generated explanations and ground truth ones (annotated by humans).

**LlaVA (Liu et al., 2023b) (instruction following)** consists of synthetically generated instructions of images from the COCO dataset. The authors use GPT-4 (OpenAI, 2023) to generate intricate instructions that can be categorized into 3 categories; 23K detailed descriptions, 77K complex questions, and 58K examples of conversations between humans and an AI agent. To generate the instruction, GPT-4 (text-only) is prompted with several handcrafted examples (ICL). To make it understand images, the image is transformed into a set of bounding boxes and captions, passed as a sequence of textual tokens to GPT-4. For each category, we sample randomly some examples from the dataset of the same category. GPT-4 is used to evaluate models quantitatively (Liu et al., 2023b). Specifically, we ask text-only GPT-4 to evaluate the performance and give a an overall score. However, evaluation based on LLMs are biased and might contain some flaws.

## F ADDITIONAL EVALUATION EXPERIMENTS

### F.1 HALLUCINATION

Table 4: **Hallucinations**. Comparison with other image captioning models. *: zeros-hot without any context (in contrast to a la Flamingo used in the paper). SoTA results from (Dai et al., 2023b; Shukor et al., 2023b).

| Method | CIDEr ↑ | CHAIR$_S$ ↓ | CHAIR$_I$ ↓ |
|---|---|---|---|
| BLIP$_{Large}$ (Li et al., 2022) | 136.70 | 8.8 | 4.7 |
| VinVL$_{Larg}$ (Zhang et al., 2021) | 130.8 | 10.5 | 5.5 |
| OSCAR$_{Base}$ (Li et al., 2020) | 117.6 | 13.0 | 7.1 |
| OFA (Wang et al., 2022b) | 75.27 | 4.36 | 3.98 |
| UnIVAL (Shukor et al., 2023b) | 91.04 | 4.44 | 3.64 |
| LMMs: Zero-shot | | | |
| OFv1-9B | 65.64 | 17.38 | 14.63 |
| OFv2-3B | 73.93 | 6.85 | 6.60 |
| OFv2-3B (I) | 73.54 | 9.07 | 8.61 |
| OFv2-4B | 73.14 | 8.35 | 7.69 |
| OFv2-4B (I) | 77.89 | 7.32 | 6.58 |
| OFv2-9B | 78.10 | 7.21 | 6.63 |
| IDEFICS-9B | 63.22/40.22* | 31.42/4.95* | 28.35/5.52* |
| IDEFICS-9B (I) | 103.42/52.31* | 18.25/5.61* | 16.96/4.19* |

In Table 4, we provide a comparison with other multimodal models. Most of these models are finetuned on COCO dataset, except for OFA and UnIVAL (that use COCO only during pretraining). Despite being an order of magnitude larger, LMMs generally hallucinate more than other baseline models. This might be due mainly to training on COCO dataset and not relying on LLMs. For IDEFICS models, we noticed very high hallucination when evaluated in zero-shot a la Flamingo.

### F.2 COMPOSITIONALITY

**CREPE.** In Figure 6, we complete our evaluation on the CREPE benchamrk by adding the results for HN-Atom + HN-Comp.

**SugarCREPE.** We evaluate LMMs on SugarCREPE. Figure 7 shows that all LMMs suffer on this benchmark, revealing that previous improvements on CREPE is coming mainly from biases in the dataset, rather than acquiring compositional ability.

**Comparison between ITM and ITS.** Figure 8 provide a comparison between ITS (HN-ITS) and ITM (HN-ITM) on the CREPE benchmark. We notice that ITS is much harder than ITM with hard negatives. We also include two baselines (ITM and ITS) where the negative caption is sampled randomly from the COCO dataset. Without hard negatives, LMMs perform very well, revealing

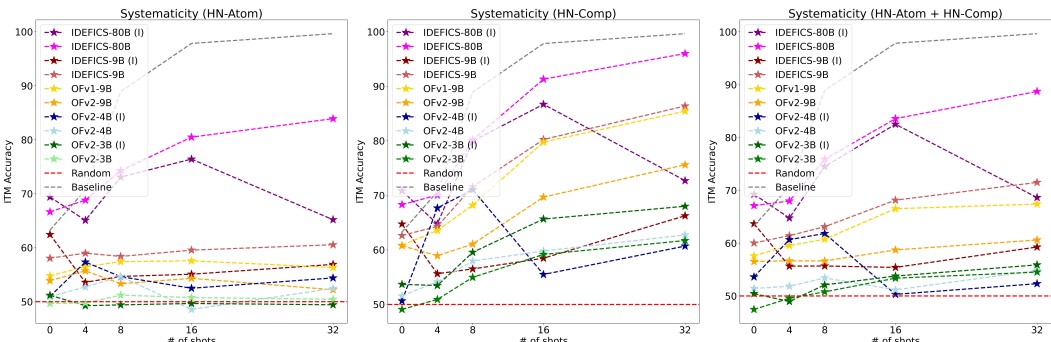

Figure 6: **Compositionality**. Models are evaluated on the CREPE benchmark with the ITM task.

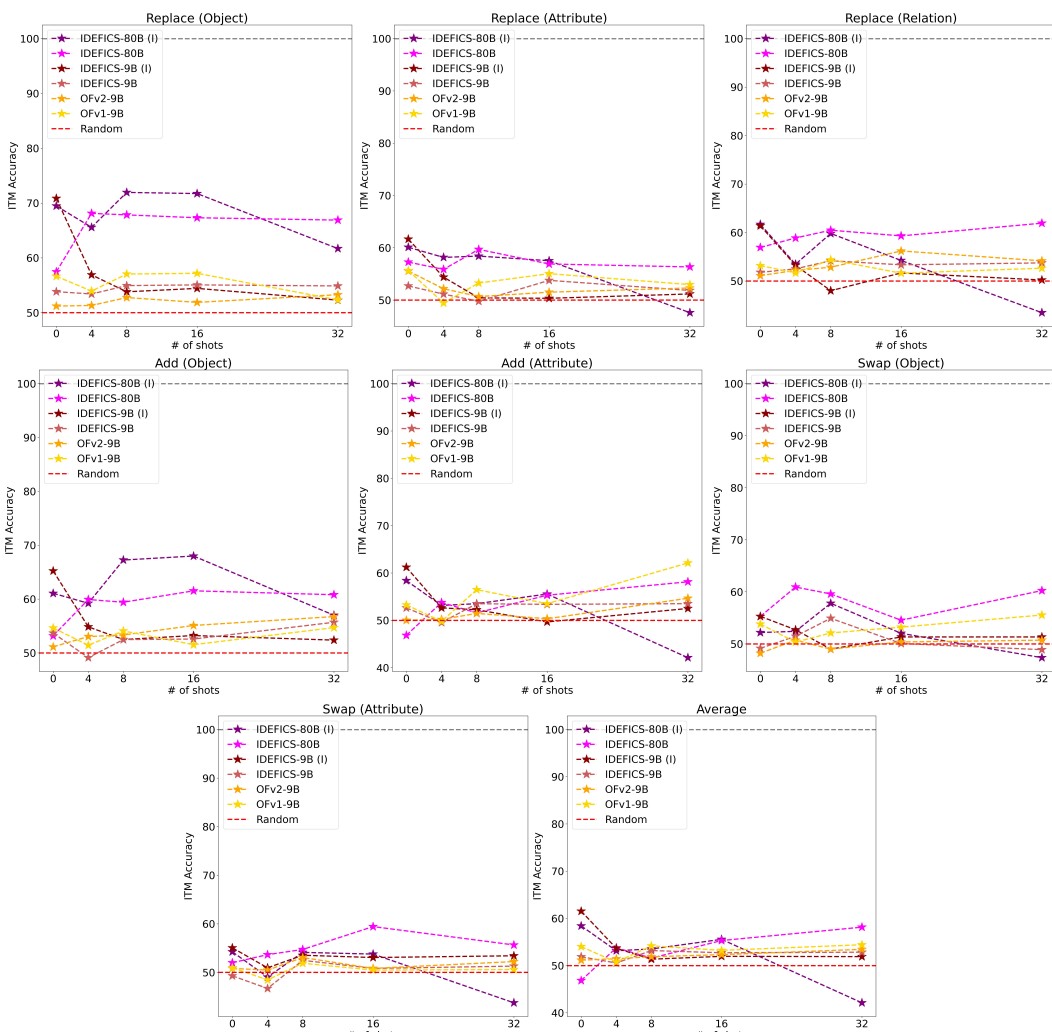

Figure 7: **Compositionality**. Models are evaluated on the SugarCREPE benchmark with the ITM task.

that the poor results with (HN-ITM/ITS) are mostly due to a lack of compositionality and not the difficulty of the task itself.

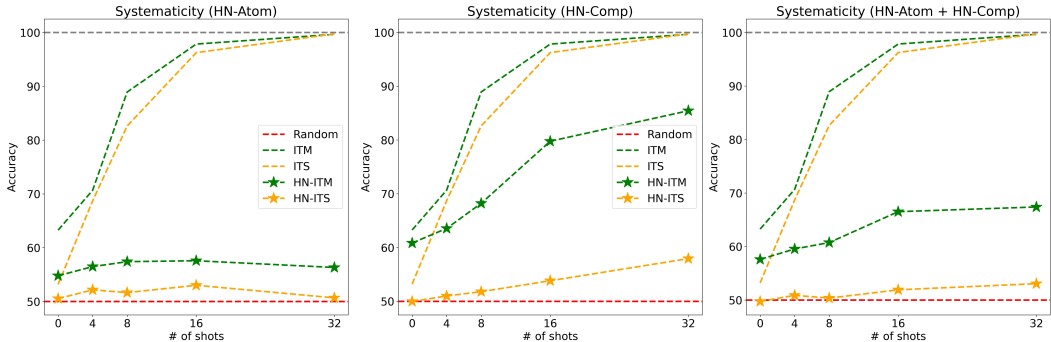

Figure 8: **Compositinality**. Comparison between ITM and ITS on the CREPE benchmark.

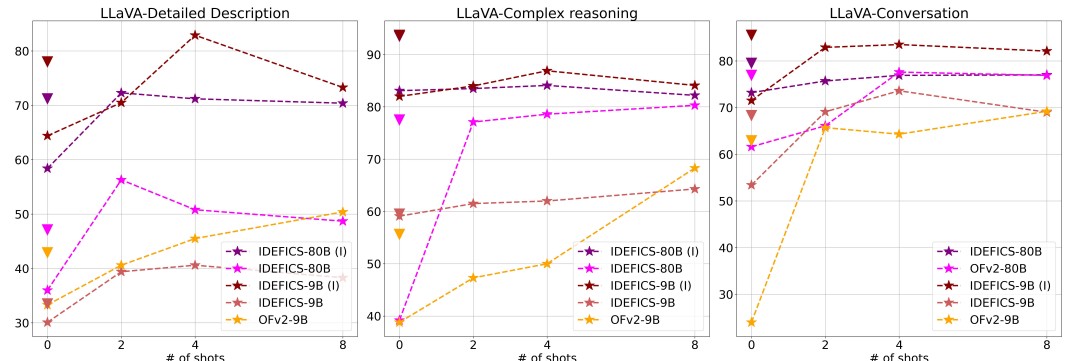

Figure 9: **Instruction following**. Quantitative evaluation on the the LlaVA benchmark on 3 types of instructions (from left to right): detailed descriptions, complex questions and conversations. ▽: 2-shots without images

## F.3 INSTRUCTION FOLLOWING

We provide additional quantitative Figure 9 and qualitative results for instruction following; detailed descriptions (Figure 11), answering complex questions (Figure 12), and conversation with humans (Figure 13) from the LlaVA benchmark. Discussion about the limitations can be found in Appendix J.

Table 5: **Mean (AVG) and Standard deviation (STD).**. We show that STD of our evaluation is not significant.

| Model | Task | | 0-shot | 4-shot | 8-shot | 16-shot | 32-shot |
|---|---|---|---|---|---|---|---|
| | OH (COCO) | AVG | 78.10/7.21/6.63 | 87.43/5.02/4.15 | 96.29/6.93/5.28 | 98.69/7.99/6.05 | 99.55/9.00/6.70 |
| | CIDEr/CHAIR$_s$/CHAIR$_i$ | STD | 0.59/0.38/0.22 | 0.22/0.29/0.13 | 0.38/0.23/0.16 | | 0.85/0.11/0.08 |
| OFv2-9B | Abstention (VQA-X) | AVG | 40.17/73.23/29.02 | 40.93/73.46/28.27 | 44.71/75.50/42.02 | 46.83/77.84/51.80 | 46.63/79.13/56.44 |
| | Acc/Absurd Acc/Absurd F1 | STD | 0.38/0.39/0.45 | 0.29/0.82/1.13 | –/0.61/0.99 | 0.35/0.30/0.31 | 0.46/0.34/0.51 |
| | Compositionality (CREPE) | AVG | 53.88/60.75/56.53 | 55.70/ 58.93/56.64 | 53.32/61.06/56.63 | 54.32/69.67/58.71 | 52.20/75.61/60.59 |
| | HN-Atom/HN-Comp/HN-Atom+Comp | STD | 0.32/0.29/0.93 | 0.86/0.62/0.64 | 0.62/0.65/0.38 | 0.40/0.58/0.82 | 0.43/0.16/0.19 |
| | Explainability (VQA-X) | AVG | 56.17 | 61.43 | 74.71 | 80.41 | 80.51 |
| | CIDEr | STD | 1.15 | 0.98 | 2.94 | 1.53 | 2.04 |
| | OH (COCO) | AVG | 40.2237/4.95/5.52 | 100.54/9.39/6.96 | 102.15/9.27/6.81 | 102.19/9.37/6.88 | 103.18/9.56/6.99 |
| | CIDEr/CHAIR$_s$/CHAIR$_i$ | STD | 0.55/0.25/0.31 | 0.73/0.04/0.06 | 0.49/0.15/0.04 | 0.24/0.21/0.12 | 0.40/0.36/0.26 |
| IDEFICS-9B | Abstention (VQA-X) | AVG | 42.82/73.85/26.87 | 45.41/74.73/32.00 | 51.89/77.12/47.51 | 58.01/80.39/60.22 | 61.94/81.75/67.45 |
| | Acc/Absurd Acc/Absurd F1 | STD | 0.27/0.60/0.90 | 0.49/0.11/0.19 | 0.20/0.54/0.75 | 0.28/0.15/0.31 | 0.59/0.18/0.52 |
| | Compositionality (CREPE) | AVG | 58.05/62.63/60.02 | 58.98/64.31/61.41 | 58.38/71.59/63.11 | 59.54/ 80.24/68.13 | 60.54/86.39/71.49 |
| | HN-Atom/HN-Comp/HN-Atom+Comp | STD | 0.72/0.44/0.35 | 0.42/0.52/0.54 | 0.26/0.44/0.35 | 0.12/0.40/0.47 | 0.49/0.44/0.22 |
| | Explainability (VQA-X) | AVG | 67.43 | 80.13 | 83.45 | 88.59 | 90.18 |
| | CIDEr | STD | 0.56 | 0.31 | 1.60 | 1.15 | 2.95 |

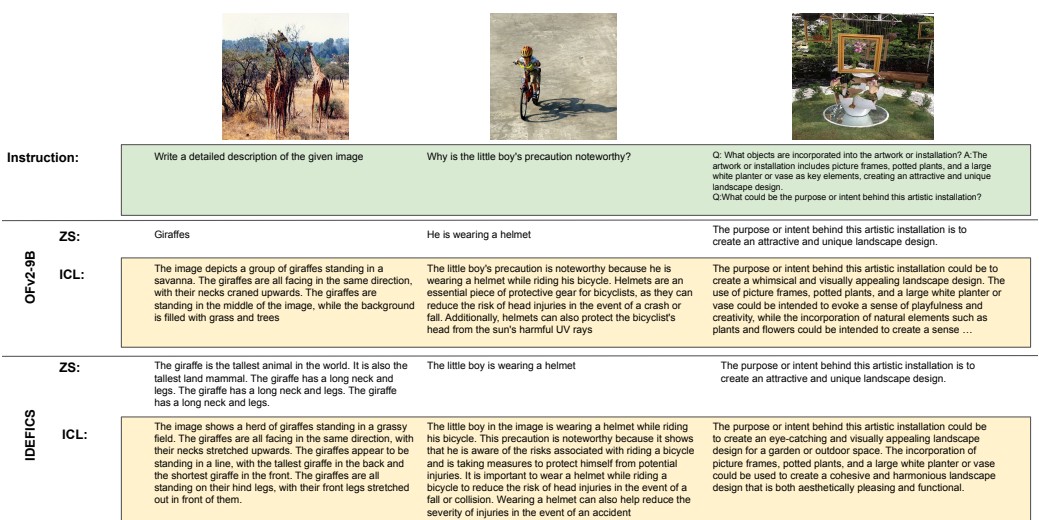

Figure 10: **Instruction following.** Qualitative evaluation results of IDEFICS and OFv2-9B on the LlaVA benchmark on 3 types of instructions (from left to right): detailed descriptions, complex questions and conversations.

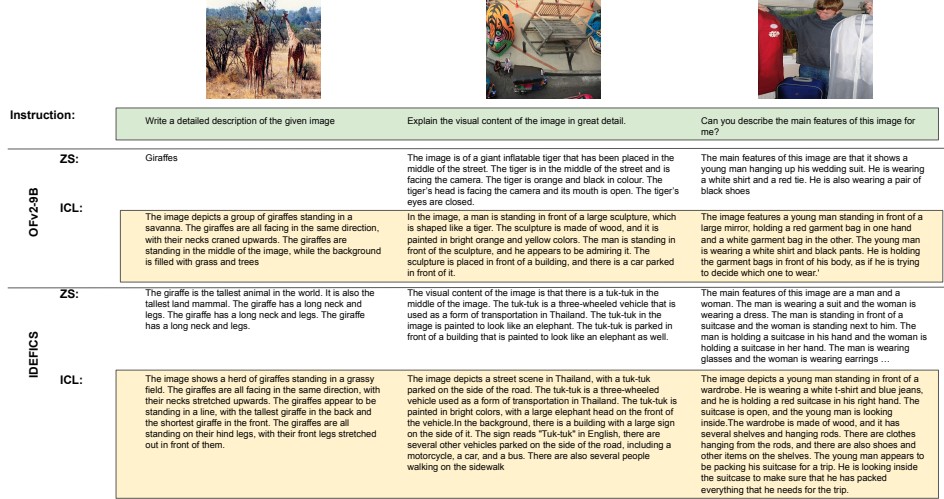

Figure 11: Illustration of instruction following on detailed descriptions.

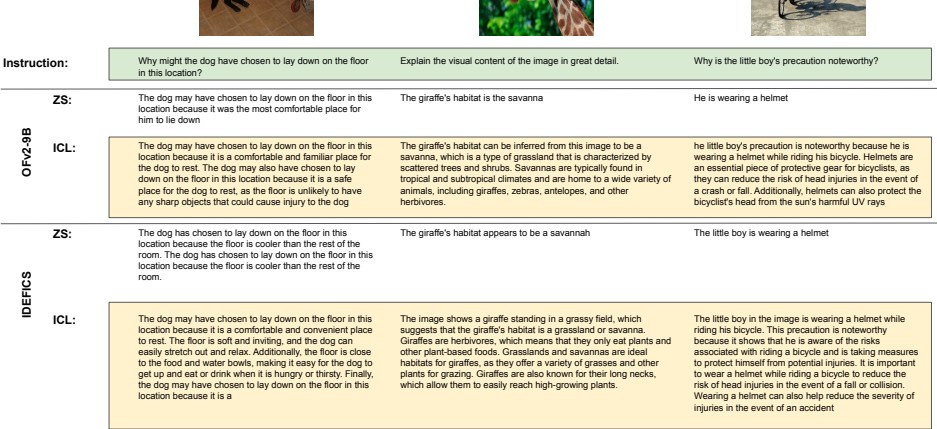

Figure 12: Illustration of instruction following on complex questions.

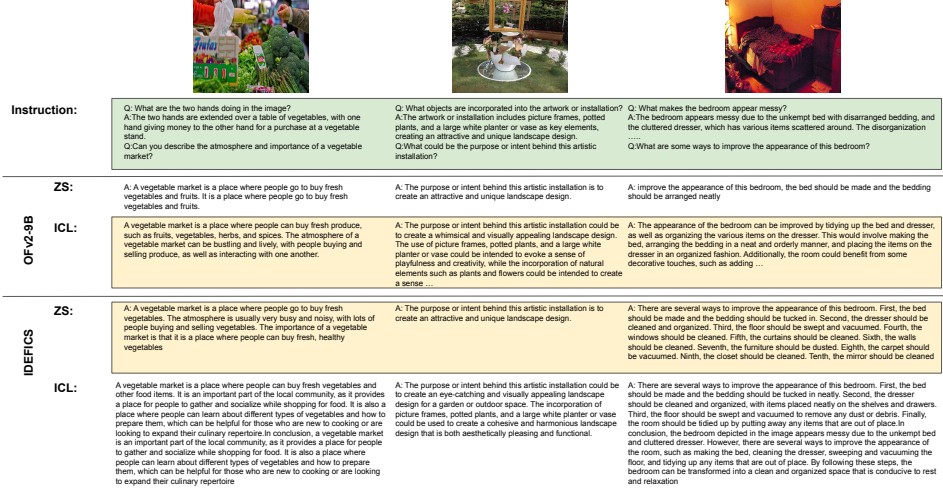

Figure 13: Illustration of instruction following on conversations.

## G ADDITIONAL DETAILS FOR X-ICL

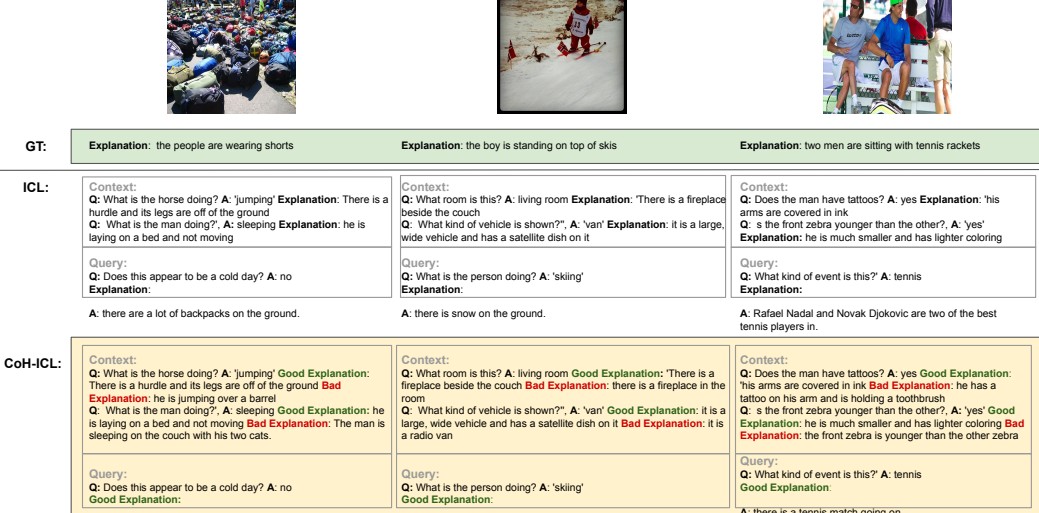

Figure 14: Explainability with CoH-ICL. The model is prompted with good (written by humans) and bad explanations (from previous model generations).

**Chain-of-Hindsight ICL (CoH-ICL).** Chain of Hindsight (CoH) (Liu et al., 2023a) consists of training the model to generate both helpful and unhelpful answers, by providing both answers as input to the LLM, each preceded by a the corresponding prompt (*e.g.* "helpful answer:", "unhelpful answer:"). Inspired by this, we propose CoH-ICL. Specifically, for each image we collect a positive and negative description. The good/positive description (image caption) is annotated by humans and the bad/negative description is generated by the model itself. As illustrated in Figure 14, during ICL the context consists of several examples as follows; an image, question, answer, human annotation as the good response, and previous model's generation (with ICL 32-shot) as the bad response. More formally, Equation (1) for CoH-ICL can be written as:

$$C = \{\langle \texttt{<image>}_i T_i T_i^+ R_i^+ T_i^- R_i^- \texttt{<|endofchunk|>}\rangle\}_N \text{ and } I = \langle \texttt{<image>} TT^+ \rangle. \quad (5)$$

where $T^+/R^+$ and $T^-/R^-$ refer to positive and negative demonstrations respectively.

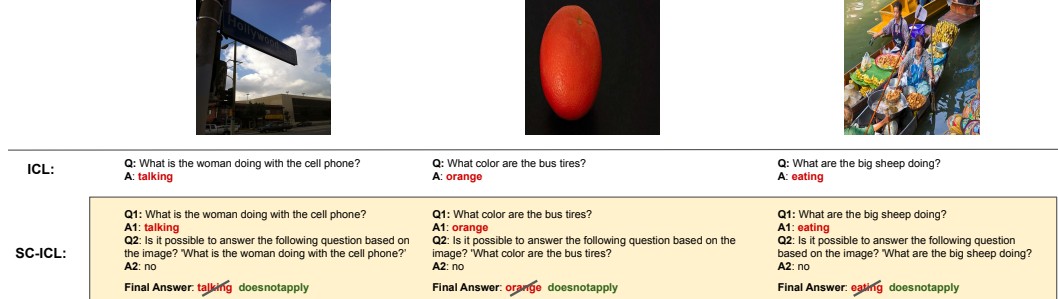

Figure 15: Illustration of SC-ICL for answer abstention.

**Self-Correcting ICL (SC-ICL).** Self-correction (SC) (Pan et al., 2023; Madaan et al., 2023; Raunak et al., 2023), consists of using the model itself to automatically correct its generated answers. We explore similar approach to help the model abstain from answering. As illustrated in Figure 15, our SC-ICL consists of the following steps:

1. We first simply ask the model the question Q using ICL, and the model gives an answer A. This is the typical ICL approach used to evaluate the model on different VQA benchmarks.

2. Then, we provide the same question Q as input and ask the model if it is relevant or answerable given the image.

3. In case the model recognizes that the question Q is not answerable, the previous answer A is ignored and replaced with an abstention keyword. Note that, in case of SC, usually the model itself correct the answers, but here we employ this heuristic mechanism.

Formally, the SC-ICL steps can be compressed in 2 steps as follows:

$$C_1 = \{\langle \texttt{<image>}_i T_i R_i \texttt{<|endofchunk|>}\rangle\}_N, I_1 = \langle \texttt{<image>} T\rangle, o_1 = LMM([C_1, I_1]), \qquad (6)$$

$$C_2 = \{\langle \texttt{<image>}_i T^{2"} T_i" R^2 \texttt{<|endofchunk|>}\rangle\}_N, I_2 = \langle \texttt{<image>} T^{2"} T"\rangle, o_2 = LMM([C_2, I_2]),$$

where $T^2$ is a fixed question to ask the model if the following question $T_i$ is relevant to the image, and $R^2$ is yes or no. The final answer is given as a function $F$ (heuristics) of $o_1$ and $o_2$, i.e., $o = F(o_1, o_2)$.

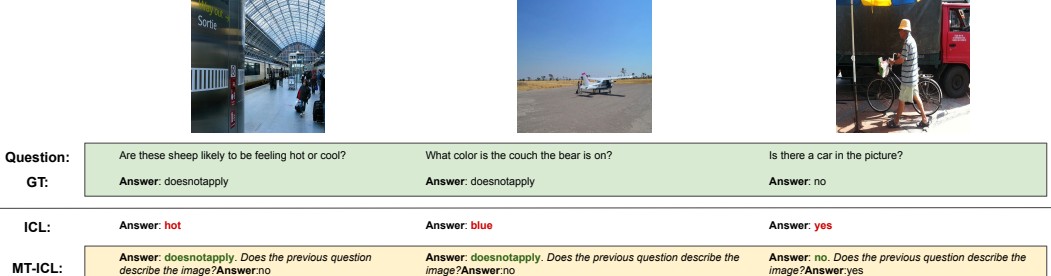

Figure 16: Illustration of MT-ICL for answer abstention.

**Multitask ICL (MT-ICL).** Multitask learning (Caruana, 1997) consists of training the same model on different tasks. We propose to do multitask learning in context. Specifically, the demonstrations contain two tasks, such as simultaneously answering the question and deciding whether the question is answerable or not. Figure 16 illustrate MT-ICL for model abstention. More formally, With $T_i^j R_i^j$ referring to task $j$, the context $C$ in Equation (1) for MT-ICL can be written as:

$$C = \{\langle \texttt{<image>}_i T_i^1 R_i^1 T_i^2 R_i^2 \texttt{<|endofchunk|>}\rangle\}_N \text{ and } I = \langle \texttt{<image>} T^1\rangle. \qquad (7)$$

# H ADDITIONAL X-ICL EXPERIMENTS

Here we provide additional experiments with different X-ICL variants to address hallucinations, abstention, compositionality, and explainability. We skip the instruction following ability as we do not have quantitative metrics to measure the improvements over ICL.

## H.1 EXPLAINABILITY

Table 6: **Explainability**. Overall task accuracy and CIDEr for explanations on VQA-X. ICL here refers to single task ICL (answer or explain).

| Model | Method | Acc. \| CIDEr | | | | | | | |
|---|---|---|---|---|---|---|---|---|---|
| | | 4-shot | | 8-shot | | 16-shot | | 32-shot | |
| OFv1-9B | ICL | 64.07 | 67.41 | 67.03 | 74.52 | 69.68 | 80.53 | 71.21 | 84.1 |
| | CoH-ICL | – | 76.43 (+9.02) | – | 80.48 (+5.96) | – | 83.15 (+2.62) | – | 87.29 (+3.19) |
| | MT-ICL | 66.02 (+1.95) | 71.75 (+4.34) | 70.06 (+3.03) | 73.2 (-1.32) | 72.07 (+2.39) | 77.89 (-2.64) | 73.22 (+2.1) | 79.23 (-4.87) |
| OFv2-9B | ICL | 69.52 | 61.43 | 72.71 | 74.71 | 73.11 | 80.41 | 72.93 | 80.51 |
| | CoH-ICL | – | 70.76 (+9.33) | – | 78.97 (+4.26) | – | 82.27 (+1.86) | – | 73.22 (-6.29) |
| | MT-ICL | 74.16 (+5.64) | 67.62 (+6.19) | 75.79 (+3.08) | 74.88 (+0.17) | 74.89 (+0.78) | 77.24 (-3.83) | 74.42 (+2.49) | 76.40 (-4.09) |
| IDEFICS-9B | ICL | 74.63 | 80.13 | 75.30 | 83.45 | 76.12 | 88.59 | 76.03 | 90.18 |
| | CoH-ICL | – | 82.21 (+2.08) | – | 86.85 (+3.40) | – | 89.00 (+0.41) | – | 92.18 (+2.00) |
| | MT-ICL | 74.80 (+0.17) | 81.06 (+0.93) | 76.51 (+1.21) | 83.51 (+0.06) | 76.75 (-0.63) | 83.56 (-4.56) | 78.03 (+2.0) | 85.86 (-4.32) |
| IDEFICS-9B (I) | ICL | 83.93 | 90.06 | 84.35 | 94.54 | 84.36 | 96.33 | 82.90 | 94.27 |
| | CoH-ICL | – | 94.87 (+4.81) | – | 93.58 (-1.96) | – | 95.75 (-0.42) | – | 96.32 (+2.05) |
| | MT-ICL | 78.69 (-5.24) | 94.93 (+4.87) | 80.40 (-3.95) | 100.14 (+5.60) | 81.22 (-3.14) | 103.39 (+7.06) | 82.51 (-0.39) | 104.70 (+10.43) |

**CoH-ICL.** Table 6 provides additional results with CoH-ICL. CoH-ICL significantly improves the scores over ICL with all models. We also provide some qualitative results in Figure 14 to illustrate the approach.

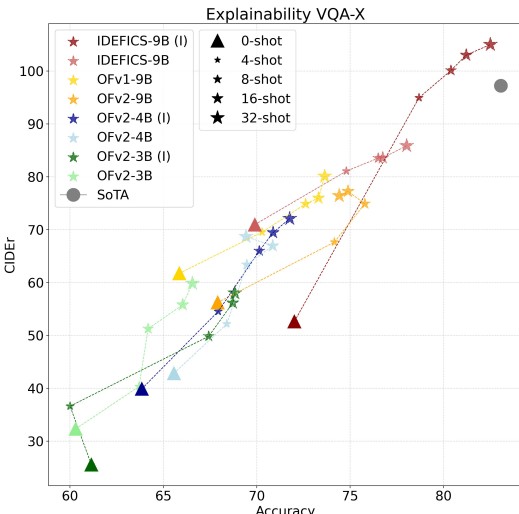

Figure 17: Explainability with MT-ICL. The model is asked to answer the question and explain its answer. We report the CIDEr (↑) for explainability and the overall VQA accuracy (↓).

**MT-ICL.** Figure 17 shows a comparison between different LMMs. LMMs answer the question and then provide an explanation. Increasing the number of shots in ICL significantly improves both tasks. Interestingly, IDEFICS (I) is able to surpass the current SoTA (NLX-GPT (Sammani et al., 2022b) unfiltered scores). In addition, Table 6 provide results for different models. Compared to ICL, the overall accuracy is increased with OFv1, OFv2, and IDEFICS models. For CIDEr, the improvement is mostly with a small number of shots, except IDEFICS (I).

## H.2 ABSTENTION

**SC-ICL.** In Table 7, we provide the results with SC-ICL (correction with the same number of shots in both SC steps) and SC-ICL (32 shots) (correction with 32 shots). SC-ICL (32shot) is significantly better than SC-ICL which is expected as classifying questions (relevant to the image or not) is better with more shots. We illustrate SC-ICL in Figure 15. With IDEFICS (I) model, the model tends to answer the question instead of deciding if it is relevant or not (which might also be the reason why the improvement margin with IDEFICS is generally smaller than OF models). More adapted prompts should fix that, which we keep for future work.

**MT-ICL.** The model here, simultaneously answers the question and decides whether the question is relevant to the image or not. Table 7 shows that MT-ICL is better than ICL on answer abstention, especially with a small number of shots. We illustrate MT-ICL in Figure 16.

Table 7: **Abstention**. We evaluate the ability the model to abstain on the TDIUC dataset.

| Model | Method | Acc. \| Absurd Acc. \| Absurd F1 | | | | | | | | | | | |
| --- | --- | --- | --- | --- | --- | --- | --- | --- | --- | --- | --- | --- | --- |
| | | 4-shot | | | 8-shot | | | 16-shot | | | 32-shot | | |
| OFv1-9B | ICL | 37.14 | 67.82 | 31.04 | 44.71 | 69.96 | 43.90 | 52.87 | 76.64 | 57.80 | 57.16 | 79.40 | 63.87 |
| | MT-ICL | 42.49 | 72.75 | 36.17 | 47.33 | 74.34 | 47.6 | 52.63 | 76.68 | 57.31 | 55.83 | 77.49 | 62.88 |
| | SC-ICL | 39.82 | 62.52 | 41.61 | 46.49 | 68.01 | 49.64 | 53.53 | 75.10 | 59.35 | 57.23 | 78.13 | 64.92 |
| | SC-ICL (32shot) | 45.34 | 70.72 | 52.00 | 50.53 | 73.30 | 57.11 | 54.98 | 76.75 | 62.46 | 57.22 | 78.10 | 64.86 |
| OFv2-9B | ICL | 40.93 | 73.46 | 28.27 | 44.71 | 75.50 | 42.02 | 46.83 | 77.84 | 51.80 | 46.63 | 79.13 | 56.44 |
| | MT-ICL | 47.99 | 77.18 | 29.99 | 48.41 | 76.98 | 48.09 | 49.13 | 76.40 | 54.58 | 48.83 | 78.09 | 59.14 |
| | SC-ICL | 43.32 | 70.93 | 42.50 | 47.26 | 72.76 | 52.57 | 47.75 | 75.56 | 56.57 | 48.25 | 77.7 | 60.16 |
| | SC-ICL (32shot) | 44.38 | 73.3 | 47.34 | 46.92 | 74.95 | 52.85 | 48.38 | 76.51 | 57.41 | 47.86 | 77.49 | 59.93 |
| IDEFICS-9B | ICL | 45.41 | 74.73 | 32.00 | 51.89 | 77.12 | 47.51 | 58.01 | 80.39 | 60.22 | 61.94 | 81.75 | 67.45 |
| | MT-ICL | 48.30 | 76.61 | 37.82 | 51.80 | 78.90 | 48.69 | 54.76 | 81.26 | 59.55 | 58.51 | 82.67 | 67.57 |
| | SC-ICL | 45.13 | 68.49 | 43.23 | 52.27 | 74.67 | 53.41 | 58.75 | 79.64 | 62.55 | 62.66 | 81.84 | 68.62 |
| | SC-ICL (32shot) | 49.56 | 77.06 | 49.56 | 54.75 | 78.89 | 57.76 | 59.21 | 80.73 | 64.16 | 62.77 | 82.01 | 68.96 |
| IDEFICS-9B (I) | ICL | 59.57 | 79.34 | 29.91 | 63.30 | 82.65 | 46.23 | 66.94 | 85.85 | 61.16 | 70.69 | 88.35 | 72.75 |
| | MT-ICL | 60.24 | 79.77 | 35.38 | 63.64 | 83.30 | 50.82 | 68.20 | 86.17 | 64.88 | 68.66 | 86.91 | 68.39 |

### H.3 OBJECT HALLUCINATIONS

**MT-ICL.** For object hallucinations, we use object recognition (listing existing objects in the image without localization) as an auxiliary task (using the prompt "There is only these objects:"). The motivation is that recognizing objects in the image might push the model to describe only seen objects. From Table 8, we noticed that this approach reduces object hallucinations when the hallucinations is significant (OFv1-9B and IDEFICS).

Table 8: **Hallucinations.** We evaluate object hallucinations on the COCO dataset.

| Model | Method | CIDEr \| CHAIR$_S$ \| CHAIR$_I$ | | | | | | | | | | |
| | | 4-shot | | | 8-shot | | | 16-shot | | | 32-shot | | |
|---|---|---|---|---|---|---|---|---|---|---|---|---|---|
| OFv1-9B | ICL | 75.36 | 13.53 | 10.82 | 78.98 | 13.78 | 10.90 | 81.38 | 13.94 | 11.08 | 83.82 | 14.08 | 11.11 |
| | MT-ICL | 73.88 | 12.38 | 10.34 | 77.60 | 12.68 | 10.49 | 80.5747 | 13.32 | 10.59 | 81.57 | 13.04 | 10.41 |
| OFv2-9B | ICL | 87.43 | 5.02 | 4.15 | 96.29 | 6.93 | 5.28 | 98.69 | 7.99 | 6.05 | 99.55 | 9.00 | 6.70 |
| | MT-ICL | 90.46 | 5.64 | 4.54 | 94.13 | 6.43 | 5.03 | 96.01 | 8.03 | 6.16 | 94.60 | 10.74 | 8.25 |
| IDEFICS-9B | ICL | 100.54 | 9.39 | 6.96 | 102.15 | 9.27 | 6.81 | 102.19 | 9.37 | 6.88 | 103.18 | 9.56 | 6.99 |
| | MT-ICL | 96.44 | 7.76 | 6.08 | 99.70 | 8.08 | 6.13 | 101.72 | 7.73 | 5.94 | 103.80 | 7.66 | 5.85 |
| IDEFICS-9B (I) | ICL | 133.89 | 3.76 | 2.56 | 136.12 | 3.90 | 2.65 | 136.81 | 3.88 | 2.62 | 136.56 | 3.89 | 2.60 |
| | MT-ICL | 129.84 | 4.79 | 3.15 | 132.55 | 4.42 | 2.96 | 134.25 | 4.36 | 2.97 | 135.99 | 3.78 | 2.62 |

### H.4 COMPOSITIONALITY

Table 9: **Compositionality.** We evaluate compositionality on the CREPE benchmark.

| Model | Method | HN-Atom \| HN-Comp \| HN-Atom + HN-Comp | | | | | | | | | | |
| | | 4-shot | | | 8-shot | | | 16-shot | | | 32-shot | | |
|---|---|---|---|---|---|---|---|---|---|---|---|---|---|
| OFv1-9B | ICL | 56.48 | 63.55 | 59.54 | 57.4 | 68.21 | 60.74 | 57.57 | 79.77 | 66.51 | 56.32 | 85.44 | 67.39 |
| | MT-ICL | 57.57 | 64.88 | 60.58 | 56.47 | 69.89 | 61.83 | 58.31 | 77.55 | 64.60 | 59.62 | 81.28 | 67.99 |
| OFv2-9B | ICL | 55.70 | 58.93 | 56.64 | 53.32 | 61.06 | 56.63 | 54.32 | 69.67 | 58.71 | 52.20 | 75.61 | 60.59 |
| | MT-ICL | 57.18 | 68.54 | 61.00 | 55.67 | 78.74 | 63.94 | 54.52 | 88.53 | 68.45 | 52.88 | 84.85 | 66.96 |
| IDEFICS-9B | ICL | 58.98 | 64.31 | 61.41 | 58.38 | 71.59 | 63.11 | 59.54 | 80.24 | 68.13 | 60.54 | 86.39 | 71.49 |
| | MT-ICL | 56.86 | 65.25 | 60.71 | 57.32 | 71.99 | 62.62 | 58.45 | 78.56 | 66.46 | 59.90 | 83.28 | 71.39 |
| IDEFICS-9B (I) | ICL | 53.58 | 55.63 | 55.64 | 54.67 | 56.50 | 55.67 | 55.08 | 58.47 | 55.40 | 56.90 | 66.25 | 59.26 |
| | MT-ICL | 56.26 | 56.92 | 56.32 | 56.26 | 59.03 | 57.10 | 58.09 | 61.68 | 58.83 | 55.13 | 58.70 | 57.18 |

**MT-ICL** Here, we also consider object detection as an auxiliary task, if the model is able to detect the objects in the image, it should be able to recognize when the caption description is false (when randomly replacing objects in the caption with atomic foils). In Table 9, MT-ICL seems to have a positive effect on HN-Comp, where the ITM accuracy is significantly improved. We notice that this approach works when the performance on compositionality is lower (OFv2 and IDEFICS (I))

## I CAN TASK INSTRUCTIONS HELP ICL?

Table 10: **Task instructions used in different benchmarks (Appendix I).**

| Benchmarks | Task instructions |
|---|---|
| Object hallucinations (COCO) | Describe the following images, do not include any object not present in the image. Here are a few illustration examples: |
| Abstention (TDIUC) | Answer the following questions about the image, give short answers, if you do not know the answer or the question is not relevant to the image say doesnotapply. Here is few illustration examples: |
| Compositionality (CREPE) | You need to find if the provided sentences accurately describe the image if the composition of the sentence does not match the image then the sentence does not describe the image. You also need to detect objects that can help you decide. Here is few illustration examples: |
| Explainability (VQA-X) | You will be given a question and answer, you need to give an explanation of the given answer based on the image. Here is few illustration examples: |

In practice, LLMs are augmented with a relatively long instruction, explicitly describing the task. In this section, we investigate if giving the model an explicit instruction (illustrated in Table 10) can help. We show the results in Table 11. We can notice that the added instructions can bring significant improvements with a small number of shots. However, when adding more demonstrations (8/16/32-shot) the effect of the instructions starts to be negligible. This is expected, as more demonstrations will help the model infer more easily the task from the context examples.

Table 11: **ICL with task instructions.** Adding explicit task instructions can help get additional improvements with a small number of ICL shots.

| Model | Task | Task Instruction | 0-shot | 4-shot | 8-shot | 16-shot | 32-shot |
|---|---|---|---|---|---|---|---|
| OFv1-9B | OH (COCO) CIDEr/CHAIR$_s$/CHAIR$_i$ | ✗ | 66.40/14.24/12.37 | 75.36/13.53/10.82 | 78.98/13.78/10.90 | 81.38/13.94/11.08 | 83.82/14.08/11.11 |
| | | ✓ | 69.82/15.37/12.57 | 75.59/15.32/11.91 | 78.76/14.9/11.69 | 81.26/14.85/11.66 | 82.88/15.26/11.89 |
| | Abstention (VQA-X) Acc/Absurd Acc/Absurd F1 | ✗ | 35.01/72.95/25.56 | 37.14/67.82/31.04 | 44.71/69.96/43.90 | 52.87/76.64/57.80 | 57.16/79.40/63.87 |
| | | ✓ | 43.68/73.6/32.15 | 40.69/62.39/38.26 | 47.62/69.68/48.25 | 53.46/75.78/57.74, | 57.34/78.45/63.36 |
| | Compositionality (CREPE) HN-Atom/HN-Comp/HN-Atom+Comp | ✗ | 54.82/60.83/57.60 | 56.48/63.55/59.54 | 57.4/68.21/60.74 | 57.57/79.77/66.51 | 56.32/85.44/67.39 |
| | | ✓ | 57.57/68.57/62.27 | 56.82/66.90/60.79 | 57.75/ 76.05/65.10 | 58.11/82.17/67.87 | 58.20/85.72/69.26 |
| | Explainability (VQA-X) CIDEr | ✗ | 59.94 | 67.41 | 74.52 | 80.53 | 84.1 |
| | | ✓ | 64.33 | 70.44 | 74.58 | 79.16 | 82.98 |

## J LIMITATIONS

### J.1 INSTRUCTION FOLLOWING WITH ICL

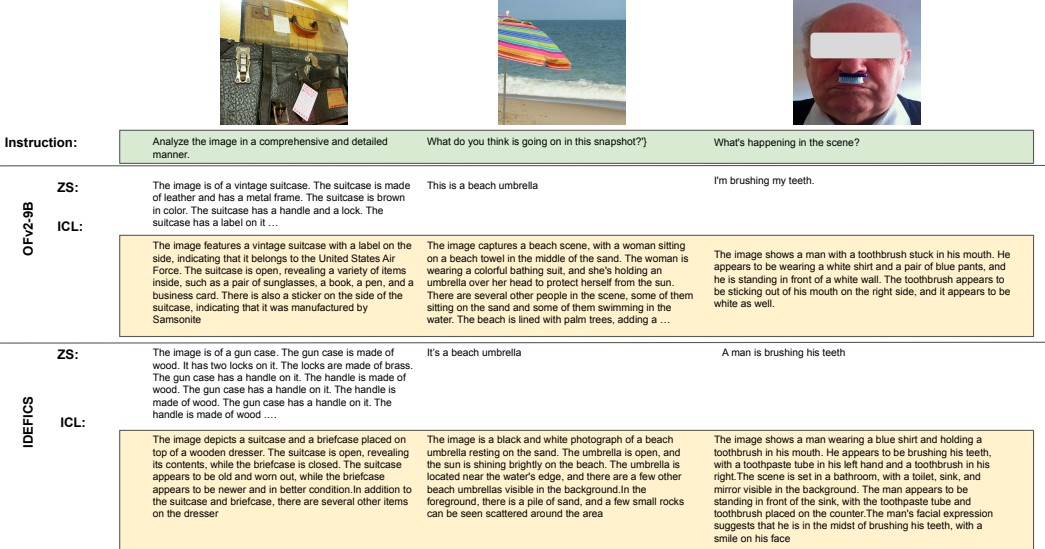

Figure 18: Limitations of instruction following with ICL.

Despite being effective in adapting the model to follow user instructions, we noticed that ICL has several limitations. Mainly, while the responses are longer and richer, they sometimes include significant hallucinations. In addition, they might include inaccurate and wrong statements that contradict what is seen in the image. Some of these limitations can be seen in Figure 18.

### J.2 X-ICL

X-ICL variants partially solve the flaws of LMMs, here we highlight X-ICL limitations to provide more inspiration for the community to devise better variants.

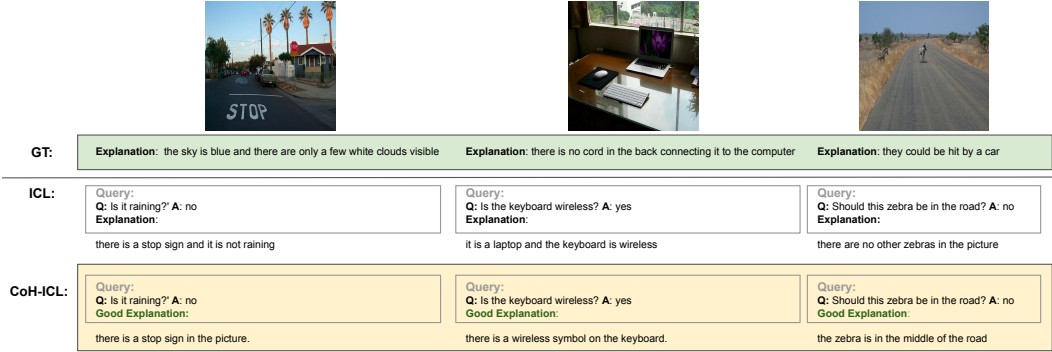

Figure 19: CoH-ICL limitations with explainability on VQA-X. The generated explanations are more like image descriptions (left), include hallucinations (middle) and be unhelpful (right).

**CoH-ICL.** This variant also suffers from several limitations as illustrated in Figure 19. In the case of explainability, the generated output is more like an image description than an actual explanation. ICL can introduce some hallucinations and provide unhelpful explanations.

**SC-ICL.** As illustrated in Figure 20, SC-ICL can fail on some abstention cases such as not recognizing the question as absurd or relevant. In addition, we correct only in case we classify the

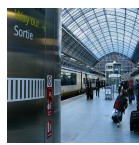 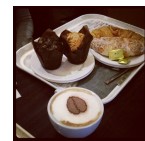 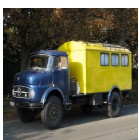

| Question: GT: | Are these sheep likely to be feeling hot or cool? **Answer**: doesnotapply | What is the table color? **Answer**: brown | What is rack made of? **Answer**: metal |
|---|---|---|---|
| ICL: | **Q**: Are these sheep likely to be feeling hot or cool? **A**: hot | **Q**: What is the table color? **A**: doesnotapply | **Q**: What is rack made of? **A**: wood |
| SC-ICL: | **Q1**: Are these sheep likely to be feeling hot or cool? **A1**: hot **Q2**: Is it possible to answer the following question based on the image? 'Are these sheep likely to be feeling hot or cool?' **A2**: yes **Final answer**: hot | **Q1**: What is the table color? **A1**: doesnotapply **Q2**: Is it possible to answer the following question based on the image? 'What is the table color? ' **A2**: no **Final answer**: doesnotapply  doesnotapply | **Q1**: What is rack made of? **A1**: wood **Q2**: Is it possible to answer the following question based on the image? 'What is rack made of? ' **A2**: no **Final answer**: wood  doesnotapply |

Figure 20: SC-ICL limitations. Some failure cases on TDIUC abstention benchmark.

question as irrelevant, thus we do not consider the case when the model abstains in step 1 and then classify the question as relevant in step 2.

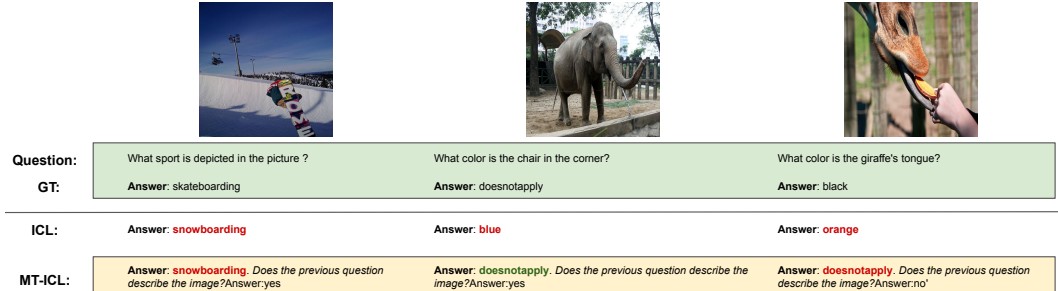

| Question: GT: | What sport is depicted in the picture ? **Answer**: skateboarding | What color is the chair in the corner? **Answer**: doesnotapply | What color is the giraffe's tongue? **Answer**: black |
|---|---|---|---|
| ICL: | **Answer**: snowboarding | **Answer**: blue | **Answer**: orange |
| MT-ICL: | **Answer**: snowboarding. *Does the previous question describe the image?*Answer:yes | **Answer**: doesnotapply. *Does the previous question describe the image?*Answer:yes | **Answer**: doesnotapply. *Does the previous question describe the image?*Answer:no' |

Figure 21: MT-ICL limitations. Some failure cases on TDIUC abstention benchmark.

**MT-ICL.** Figure 21 shows some failure cases with MT-ICL on answer abstention. Sometimes there is an inconsistency between the output of the main and auxiliary task such as replying "doesnotapply" and classifying the image as relevant. MT-ICL does not seem to help correctly respond to the question and fails to detect some abstention cases.

