# OpenReview forum: "Beyond task performance: evaluating and reducing the flaws of large multimodal models with in-context-learning"
_ICLR.cc/2024/Conference — ICLR 2024 poster_

### Official Review · Reviewer_sfnT · 2023-10-31

**Soundness:** 3 good
**Presentation:** 4 excellent
**Contribution:** 3 good
**Rating:** 8
**Confidence:** 4

**Summary:**

This paper produces a detailed analysis of various open-source Large Multimodal Models (LMMs) across 5 different axes: hallucinations, abstention, compositionality, explainability, and instruction following. The paper identifies how each model’s performance changes (or does not change) with increased ICL examples, and proposes variants of ICL prompting methods to improve performance on some of the identified categories.

**Strengths:**

The categorization of various LMM failures and the thorough analysis of how different model variants and sizes perform at each task is well done. The main takeaways from this paper, highlighting crucial limitations of current LMMs and identifying where ICL is and is not useful, and proposed methods for improving ICL in the LMM context (X-ICL) are all meaningful insights for the research community. Overall, the work seems significant, experiments are thorough and diverse, and the writing is clear.

**Weaknesses:**

- Only qualitative results are given for the Instruction Following evaluations. It would be interesting to see if other forms of automated evaluations assessing instruction following can be used (e.g. checking that detailed answers are indeed more detailed or checking whether the original question was indeed answered by passing the LMM generated text and original instruction into a LLM like GPT-4). While LLM-based evaluations can still be flawed, it can also provide a more thorough view (looking at a wider range of questions rather than the small sample that was presented qualitatively in this work). Have the authors experimented with automatic evaluations for the instruction following tasks?
- Nit: the table and figure fonts are very small and hard to see.

**Questions:**

See above question about automated evaluations for instruction following.

---

> ### Author Response · Authors · 2023-11-14
> **Response to Reviewer sfnT**
>
> We thank the reviewer for their constructive reviews, and for describing our paper as well written, significant and providing meaningful insights to the research community. In the following, we try to address the expressed concerns:
>
> ### **Rev: "Only qualitative results are given for the Instruction Following evaluations. It would be interesting to see if other forms of automated evaluations assessing instruction following can be used (e.g. checking that detailed answers are indeed more detailed or checking whether the original question was indeed answered by passing the LMM generated text and original instruction into a LLM like GPT-4). While LLM-based evaluations can still be flawed, it can also provide a more thorough view (looking at a wider range of questions rather than the small sample that was presented qualitatively in this work). Have the authors experimented with automatic evaluations for the instruction following tasks?"**
>
> We understand this concern, and agree that quantitative evaluation is indeed important to support our claims regarding Instruction following. To complete our evaluation, and as suggested by the reviewer, we conduct a quantitative evaluation with GPT-4.
>
> * Setup: for each question-instruction pair we provide 2 answers to GPT-4: one generated by our evaluated models, and another by GPT-4 itself, and ask it to provide scores for these answers. The reported scores signify the score of our model w.r.t GPT-4 (100% means the evaluated model has the same quality as GPT-4).
>
> * Results: the results in Fig.5 (in the revised paper) support our previous findings; models that are not instruction tuned (IDEFICS and OFv2-9B) struggle to follow user instructions, and ICL improves in this regard, making them more helpful.
>
> ### **Rev: "R: Nit: the table and figure fonts are very small and hard to see."**
>
> Indeed, we fix this in the revised paper

---

### Official Review · Reviewer_dAEJ · 2023-11-01

**Soundness:** 3 good
**Presentation:** 4 excellent
**Contribution:** 4 excellent
**Rating:** 8
**Confidence:** 3

**Summary:**

This study assesses the shortcomings of open-sourced LMMs (3B-9B) in areas such as object hallucination, abstention, compositionality, explainability, and instruction following. The authors show that LMMs face challenges in most areas despite having ~9B parameters. The impact of In-context Learning (ICL) on these limitations is also examined, showing mixed effects; it helps in some aspects, worsens others, or has minimal impact. Variants of ICL are proposed to mitigate some flaws, aiming to continually improve benchmarks.

**Strengths:**

1. The paper is well-written, clearly highlighting essential results. The use of colored words effectively guides the reader to the relevant axes under examination.
2. By offering novel insights, the paper contributes meaningfully to the future development of LMMs.
3. A commendable effort has been made to cover multiple open-sourced LMMs
4. The authors thoughtfully propose modifications to ICL, aiming to induce improvements along various axes.

**Weaknesses:**

1. While the modifications to ICL yield improvements in open-sourced LMMs, it would be advantageous to evaluate these findings in the context of models with >9B parameters, as these larger models are prevalent in the state-of-the-art (SOTA) and also have been shown to exhibit different trends than the smaller models.
2. A more comprehensive discussion on why longer CL (32 & 64) does not yield improvements across various axes would enhance the understanding of the models' behaviors.
3. The text in all illustration figures is very difficult to read, and Table 3 is also challenging to interpret due to the small font size.

**Questions:**

With three random trials conducted, it remains unclear whether the reported results in the tables represent mean or median values.

---

> ### Author Response · Authors · 2023-11-14
> **Response to Reviewer dAEJ**
>
> We thank the reviewer for these helpful comments, and highly appreciate finding our work; well written, thorough in terms of experiments, provide novel insight and contribute meaningfully to the community. In the following, we try to address the expressed concerns:
>
> ### **Rev: "While the modifications to ICL yield improvements in open-sourced LMMs, it would be advantageous to evaluate these findings in the context of models with >9B parameters, as these larger models are prevalent in the state-of-the-art (SOTA) and also have been shown to exhibit different trends than the smaller models."**
>
> * Very large models are indeed known to possess some emergent abilities not present in smaller models. However, models up to 9B parameters also possess such abilities and attain SoTA results on most of the benchmarks. We didn’t include larger models in the paper because of the computation cost associated with evaluating such models.
> * To answer this point, we complete our evaluation by considering the largest open source LMMs (IDEFICS-80B and IDEFICS-80B-Instruct); models up to 80B parameters (that can fit on 8GPUs A100). We update the figures in the revised paper. Our findings remain the same, with slight modifications: These models (80B) perform generally better than the 9B-parameter ones, however our claims regarding each axis remain mostly the same.
>
> * The claims that are slightly modified are as follows:
>     - Object hallucinations: 80B-parameter models also suffer from hallucinations, similar to smaller models. However, they suffer less from hallucination amplification when tested with ICL.
>     - Compositionality: seemingly 80B-parameter models perform significantly better on the CREPE benchmark. However, due to the importance of this axis, we deepen our study and consider a more challenging benchmark (SugarCREPE), with less biases to exploit (results in Appendix F.2). On this benchmark, we show that indeed, all models including 80B-parameter ones do not acquire compositional ability. Thus, our claim holds here as well.
>
> ### **Rev: "A more comprehensive discussion on why longer CL (32 & 64) does not yield improvements across various axes would enhance the understanding of the models' behaviors."**
>
> This is an interesting observation that is worth elaborating and be included in the revised paper (Appendix A in the revised paper). The performance saturation after many shots is not only seen in our study, but also in the original work of OpenFlamingo and IDEFICS. For example, in Fig.6 in the OpenFlamingo paper; the VQA accuracy  saturates or even degrades  after 4/8 shots. Similarly for IDEFICS, but slightly better  (https://huggingface.co/HuggingFaceM4/idefics-9b-instruct).
>
> There is multiple reasons for why multimodal ICL is not as effective as in LLMs, we list the following potential ones:
>    - Training data:  the multimodal datasets are still an order of magnitude smaller than those for LLMs. In addition,   the web documents used to train such models do not contain many interleaved image-text pairs (a lot less than 32), which might hinder the ability of the model to generalize to larger number of in-context demonstrations
>    - Model scale: The trainable parameters during pretraining, are relatively small (<15B), and acquiring better ICL ability might require training more parameters for more iterations.
>
> Finally, we would like to highlight the lack of in depth analysis of ICL in the context of LMMs, which we keep for future work.
>
> ### **Rev: "The text in all illustration figures is very difficult to read, and Table 3 is also challenging to interpret due to the small font size."**
>
> We enhance the readability of the figures in the revised paper, in particular Table .3, Fig.1 and Fig.5.
>
> ### **Rev: " With three random trials conducted, it remains unclear whether the reported results in the tables represent mean or median values."**
>
> The reported results are the average over 3 runs as mentioned in the Implementation details section (“We repeat each experiment 3 times and report the averaged results…”).. We first omitted adding the variance to improve the readability of the paper. Now, we added a table (Tab.5) in the appendix showing that the variance is indeed small.

---

> ### Comment · Reviewer_dAEJ · 2023-11-20
> **Response from reviewer**
>
> Thank you for conducting the necessary experiments and answering my queries. I appreciate the efforts made, especially regarding running a bigger model thereby enhancing the quality of the submission. In recognition of these improvements and the additional insights provided, I am raising the score to 8.

---

> > ### Author Response · Authors · 2023-11-20
> > **Special thanks to reviewer dAEJ**
> >
> > We thank the reviewer for taking the time to read and acknowledge our rebuttal, and highly appreciate raising the score.
> > Should there be any remaining concerns that you think need further attention, we would be happy to provide additional clarification.

---

### Official Review · Reviewer_PTKm · 2023-11-01

**Soundness:** 3 good
**Presentation:** 3 good
**Contribution:** 3 good
**Rating:** 3
**Confidence:** 4

**Summary:**

This paper proposes to study the issues of large multimodal models and proposes a way to improve their performance through in-context learning. The paper proposes an axis of different ways to evaluate models (hallucinations, abstention, compositionally, explainability and instruction following) and illustrates how multimodal models typically struggle with this, with in-context learning further hurting performance. The paper proposes a new way to do in-context learning which improves performance along each of these axises

**Strengths:**

- This paper analyzes the performance of multimodal models and illustrates a variety fo axises on how they might fail
- This paper proposes new methods to improve the in context learning performance of methods

**Weaknesses:**

- The results lack confidence intervals -- in general there is very large variance between the numbers. For instance in Table 2 and 3, there is large variance of a single method between a different number of shots which seems larger than the reported gains of the proposed method .
- It would be good to report quantitative results for each result in the paper.
- I had difficulty understanding the main differences between the new in-context learning methods presented in the paper and the performances seem minor
-  The existing evaluated models seem out of date -- it would be good to compare with models such as BLIP or LLava, MiniGPT-4 or GPT4V

**Questions:**

- Can the authors provide psuedocode for each of the in-context learning methods that are proposed?
- Why does the modification to in-context learning of presenting both positive and negative demonstrations  correspond to CoH in context learning?

---

> ### Author Response · Authors · 2023-11-14
> **Response to Reviewer PTKm**
>
> We thank the reviewer for their time and effort in reviewing the paper, and try to address their concerns in the following:
>
> ### **Main contributions clarification:**
>
> As it seems, many of the reviewer's concerns are related to the last part of the paper (Section 3 X-ICL, Tab.2 and Tab.3), we ask the reviewer to refer to the general response to clarify more the framing and the main messages of the paper.
>
> ### **Rev: "The results lack confidence intervals -- in general there is very large variance between the numbers. For instance in Table 2 and 3, there is large variance of a single method between a different number of shots which seems larger than the reported gains of the proposed method."**
>
> First, we didn’t report the small confidence intervals to improve the readability of the graphs. As requested, we now add the standard deviation in the revised paper (Tab.5 in Appendix) for 2 representative models. Due to  evaluating on a sufficiently large number of examples (e.g. 8K for abstention) The variance is small, for example:
>
> * For OH with IDEFICS-9B 16-shots the CIDEr/CHs/CHi are 102.19/9.37/6.88 (avg) and 0.24/0.21/0.12 (std),
> * Similarly for OFv2-9B Abstention the Acc/Absurd Acc/Absurd F1 are  46.83/77.84/51.80 (avg) and 0.35/0.03/0.31 (std).
>
> This reveals that our scores are stable. In addition, the variance is very small compared to the change in scores for different numbers of shots.  This further supports our claims and findings in the paper.
>
> Regarding the variance across the number of shots; comparing different methods, and comparing the same method across different numbers of shots are two different things and should not affect the interpretation of the significance of our approach, as our goal is to compare X-ICL to ICL for a fixed number of demonstrations.
>
> ### **Rev: "It would be good to report quantitative results for each result in the paper."**
>
> * We  agree with the reviewer on the importance of providing quantitative results for all benchmarks. In the paper we initially provide scores for all models/benchmarks, except for Instruction following, due to the complexity of evaluating this task quantitatively. To complete our evaluation we now provide scores also for this task automatically by using the now available GPT-4.
>
> * The setup is as follows; for each question-instruction pair we provide 2 answers to GPT-4: one generated by our evaluated models, and another by GPT-4 itself, and ask it to provide scores for these responses. The reported scores signify the score of our model w.r.t GPT-4 (the higher the better).
>
> * The results in Fig.5 (in the revised paper) support our previous findings; models that are not instruction tuned (IDEFICS and OFv2-9B) struggle to follow instructions, and ICL improves in this regard, making them more helpful.
>
> ### **Rev: "I had difficulty understanding the main differences between the new in-context learning methods presented in the paper and the performances seem minor"**
>
> * To prioritize more important messages (Section 2), we provide more thorough discussions and details about the evaluation of LMMs and the study of ICL. As for the last part (X-ICL) and due to space limitation, we provide more explanations about different ICL variants in Appendix G.
>
> * Regarding the minor improvements on some benchmarks, we would like to emphasize that X-ICL is first without training (almost not cost) and second, very simple to implement (no complex procedure or tedious prompt engineering). With such simple methods we are able to show significant improvements on some benchmarks. Improving the performance might require more complex ICL variants which we keep for future work.

---

> ### Author Response · Authors · 2023-11-14
> **Response to Reviewer PTKm (Part 2)**
>
> ### **Rev: "The existing evaluated models seem out of date -- it would be good to compare with models such as BLIP or LLava, MiniGPT-4 or GPT4V"**
>
> * We kindly disagree with the reviewer considering the evaluated models outdated, OpenFlamingo and IDEFICS are very recent and not even published yet (IDEFICS will be published in Neurips 2023 and OpenFlamingo still a preprint).
>
> * Besides, our choice is not just based on model size, but rather on the ICL ability of the models. As far as we know (before the submission deadline) OF and IDEFICS are the best open-source models with good ICL ability, which is not the case for BLIP and LlaVA.   Specifically, BLIP and LlaVA are mostly trained on image-text pairs and pass a significant number of tokens to the input of the LLMs, which limit the ability to do ICL, as well as to do ICL with a large number of shots (due to the LLM context limit).
>
> * To try to address the reviewer's concerns, we instead  complete our pool of models with the recent open-source IDEFICS-80B-instruct (as well as IDEFICS-80B). This model was introduced after the OBELICS [1] and behaves well with ICL. The evaluation of these models support further our findings (please also check the answer to rev dAEJ).
>
> * The proposed work and the code that will be released  are meant to facilitate the evaluation of future LMMs on current and new benchmarks.
>
> [2] Laurençon, Hugo, et al. "OBELICS: An Open Web-Scale Filtered Dataset of Interleaved Image-Text Documents." NeurIPS 2023.
>
>
> ### **Rev: "Can the authors provide pseudocode for each of the in-context learning methods that are proposed?"**
>
> * The authors will release the full code to reproduce the results. As for now, we provide a link to the full code here: https://anonymous.4open.science/r/EvALign-ICL-8DC9/README.md. The different ICL variants can be found in open_flamingo/eval/ in the following files; caption_utils.py/vqa_utils.py and itm_utils.py
>
> * In addition, we provide the necessary details to implement these variants in the paper in form of equations (eq. 2 and 3), illustration figures (Fig. 13/15/16), in addition to more details in Appendix G. For ICL, we follow the typical pipeline used in IDEFICS and openflamingo, that you can also check in https://github.com/mlfoundations/open_flamingo
>
> ### **Rev: "Why does the modification to in-context learning of presenting both positive and negative demonstrations correspond to CoH in context learning?"**
>
> This is inspired by CoH [2], in which, the authors pass **during training** both positive and negative examples to the model as input (instead of passing only the positive one).  In our case, we do the same, but instead of training the model, we do it during ICL. This shows that those approaches, first introduced for training, can actually work at test time with ICL, thus improving efficiency and practicality.
>
> [2] Liu, Hao, Carmelo Sferrazza, and Pieter Abbeel. "Languages are rewards: Hindsight finetuning using human feedback." arXiv 2023.

---

> ### Author Response · Authors · 2023-11-20
>
> Dear Reviewer, before the discussion period ends, we would love to know if you had the time to read our rebuttal, and whether additional clarification is required. Thank you again for reviewing our work

---

> ### Comment · Reviewer_PTKm · 2023-11-28
> **Rebuttal Response**
>
> I thank the authors for their detailed rebuttal response -- however I don't think my main concerns are addressed by the rebuttal.
>
> - I would like to see studies on more well known multi-modal LLMs, for instance LLava, Mini-gpt4, or perhaps the recently released GPT-4V model.
> - I remain very concerned about the numbers reported in both Table 2 and 3 in the paper. In the original submission of the paper, the variance between identical methods across shots was very high -- calling into question the confidence interval of the results. It seems like the numbers in the revised submission might have changed and the look more consistent now -- but the authors didn't mention changing anything in the results which is a bit concerning. Looking  at Table 3, the plus and minus green/red delta numbers now don't match the actual values reported (for instance 32-shot Abst F1)
> - I also don't really understand why ICL through the hindsight alignment really would help the performance of VLMs -- given there inability to really process image/text effectively in the earlier settings. Also, wouldn't this alignment with 32 shots be very computationally expensive / intractable with context length? How did the authors deal with that?
>
> I would like to defend my current score for the submission

---

### Official Review · Reviewer_wHgE · 2023-11-06

**Soundness:** 3 good
**Presentation:** 2 fair
**Contribution:** 2 fair
**Rating:** 3
**Confidence:** 3

**Summary:**

The paper presents a series of experiments assessing Large Multimodal Models (LMMs) in five areas: object hallucinations, abstention, compositionality, explainability, and instruction following. It shows that scaling LMMs doesn't fully address their deficiencies. In-Context Learning (ICL) is investigated as a remedial measure, with mixed results: it helps in some areas but not others, and can even increase hallucinations. Despite these insights, the paper suggests innovative ICL variants with potential for improvement.

**Strengths:**

The paper provides a thorough critique of LMMs by examining their performance beyond standard benchmarks and introducing new ICL approaches to overcome their limitations. It show some insight for understanding LMMs' applicability.

**Weaknesses:**

While the paper addresses an essential aspect of LMMs' functionality, it falls short in pioneering novelty. It does not introduce new datasets or unique evaluation methodologies, which limits its contribution to the field. The potential impact of the proposed ICL variants is not sufficiently analyzed, and the paper overlooks a discussion on the selection process for demonstrations in ICL. This lack of methodological detail, particularly when compared to existing demonstration selection strategies based on similarity, raises concerns about the practical implementation and reproducibility of the results.

**Questions:**

1. How is the selection process for demonstrations in context $C$ determined?
1. Could you specify the evaluation metrics used for each of the five axes and explain why the metrics can measure the axes?

---

> ### Author Response · Authors · 2023-11-14
> **Response to reviewer wHgE**
>
> We thank the reviewer for their time and effort, and try to address the expressed concerns:
>
> ### **Rev: "While the paper addresses an essential aspect of LMMs' functionality, it falls short in pioneering novelty. It does not introduce new datasets or unique evaluation methodologies, which limits its contribution to the field."**
>
> Our evaluation is novel in multiple ways:
> * The first work that highlights the limitations of LMMs from 3B to 9B (and now even 80B in the revised paper, see R.dAEJ)  on different and very important aspects, beyond mere accuracy on academic benchmarks.
> * The first work studying multimodal-ICL and its effect on addressing the limitations of LMMs, which provide a critical analysis of a de facto approach used to adapt LMMs and LLMs.
> * The first work proposing new multimodal-ICL variants that bring improvements on several benchmarks, despite their simplicity.
> We believe that such work and effort is important for the community to start focusing on fixing and addressing such limitations, instead of just saturating academic benchmarks.
>
>
> ### **Rev: "The potential impact of the proposed ICL variants is not sufficiently analyzed"**
>
> We prioritize other and more important messages in the paper, and due to lack of space we put more details about X-iCL in Appendix G. Please also check the general response to see our additional analysis and more details about how we frame the paper and which messages are important to convey and study.
>
> The main objective of adding this section at the end of the paper  (Sec.3 X-ICL) is to open up a potential direction to work on aligning and alleviating the limitations of LMMs more efficiently (as compared to training) via prompting and ICL. We keep more thorough study about X-ICL as well as more complex variants for future work.
>
>
> ### **Rev: "The paper overlooks a discussion on the selection process for demonstrations in ICL. This lack of methodological detail, particularly when compared to existing demonstration selection strategies based on similarity, raises concerns about the practical implementation and reproducibility of the results."**
>
> We kindly disagree with the reviewer on this point, and would like to clarify that the selection of the demonstrations in our ICL and X-ICL is the same; all randomly selected from the corresponding datasets, as highlighted in several places in the paper (e.g. “we follow the standard way and randomly select the demonstration examples” (Implementation details), “demonstrations are selected randomly” Sec.2.5).
>
> However, the difference between these variants is not  how we select the demonstrations, but rather, how these demonstrations are written and framed, which are illustrated in the Equations (eq 2 and 3) and figures (Fig. 13/15/16). Approaches based on picking demonstrations based on similarity such as RICES [1] are orthogonal to our ICL variants, as “which” demonstration to pick from the dataset (e.g. most similar to the query)  and “what/How” these demonstrations are passed to the model  (e.g. answers+explanations or positive+negative answers) are two different but complementary family of approaches.
>
> [1] Yang, Zhengyuan, et al. "An empirical study of gpt-3 for few-shot knowledge-based vqa."AAAI 2022.
>
>
> ### **Rev: "How is the selection process for demonstrations in context determined?"**
>
> The demonstrations are selected randomly as explained for instance in Section 2 (Implementation detail), Sec. 2.5, Appendix D
> Our selection also follows the standard setup in IDEFICS and Open Flamingo.
>
>
> ### **Rev: "Could you specify the evaluation metrics used for each of the five axes and explain why the metrics can measure the axes?"**
>
> We use standard and well-established metrics for each axis, as well as citations for each paper that explain why the underlying metric measures a given axis. The metrics for each benchmark/dataset are detailed in the “Benchmark” paragraph of each subsection of Section 2. For example, in Sec.2.1, we state: “we report the CHAIRs metric (Rohrbach et al., 2018) comparing the objects referred in the generated captioning to those actually in the image”. Due to space constraint, those metrics are further described in details in Appendix. E.

---

> > ### Author Response · Authors · 2023-11-20
> >
> > Dear Reviewer, before the discussion period ends, we would love to know if you had the time to read our rebuttal, and whether additional clarification is required. Thank you again for reviewing our work

---

### Author Response · Authors · 2023-11-14
**General response to all reviewers and paper updates**

We would like to thank all reviewers for their thoughtful and constructive comments and questions. In the following, we clarify the framing of the paper, as some of the reviewers might have missed some major messages. We also add the details for additional experiments requested by some of the reviewers (Rev. PTKm, dAEJ, and sfnT). These experiments further support our previous findings and claims.

### **Paper framing and main messages:**


Here are the main messages of the paper, in decreasing order according to their importance and their presence in our work:


* *Large-scale training is not enough*: the current trend in the multimodal community is to train larger models on larger and more diverse datasets. This is motivated by observing an accuracy increase in several academic benchmarks such as VQAv2 and COCO captioning. However, relying on mere accuracy is not enough as shown by our intricate evaluation. To this end, our work provides a critical evaluation of the recent LMMs (from 3B to 80B parameters) showing that ”LMMs have flaws that remain unsolved with scaling alone”(Abstract/Introduction) such as for hallucination (“simply scaling LMMs is not enough to reduce hallucinations” (Sec.2) and compositionality “despite scaling the number of model parameters and of training examples, LMMs still lack compositional abilities” (Sec.2).

* *Be careful when using ICL, it is still limited in many aspects*: we also provide a critical study of ICL showing where it is effective (“ICL helps them abstain” Finding.2) and where rather it hurts (“A small number of ICL shots partially alleviate it, while increasing them exacerbates the problem” Finding.1). This provides meaningful insights for the community regarding the limitations of ICL and hopefully attracts more efforts towards analyzing and understanding ICL

* *Some simple and very light ICL variants can bring additional improvements (Sec. 3*): As a preliminary study regarding point 2, we show that with simple and light ICL approaches, that are usually used during training (multitask learning, CoH learning) we can have additional gain on many axes.

* *Towards multimodal models aligned to human preferences*: while alignment to human preferences is heavily investigated for LLMs, little to no work tries to consider LMMs. We choose some of our evaluation axes in this regard (“Object hallucinations (OH) honest, harmless) ..”  in Introduction) such as hallucinations (truthfulness), abstention (honesty and reliability) and instruction following (helpfulness). We that our work “offer promising direction towards aligning efficiently them to human preferences and expectations” (Discussion)

We believe that these messages are important for the multimodal and the broader LLMs communities.

### **Paper updates:**

As requested by some reviewers we updated our paper with the following experiments, insights and information (highlighted in yellow in the revised paper):

* Adding the largest open-source models to the evaluation list (**Rev.dAEJ**): we evaluate **IDEFICS-80B** and **IDEFICS-80B-instruct** (Fig 2/3/4/5). The results support our previous findings with slight modifications.

* Quantitative evaluation for all axes (Fig.5) (**Rev.sfnT and PTKm**): we added a **quantitative evaluation based on GPT-4** for the remaining Instruction Following. This confirms our previous qualitative findings.

* We add the **variance (standard deviation)** to the appendix (Tab.5), and show that they are indeed very small (**Rev.PTKm and dAEJ**)

* Improving the fonts and the **readability of the figures** (**Rev.dAEJ and sfnT**)

* We provide an anonymous link to our code (https://anonymous.4open.science/r/EvALign-ICL-8DC9/README.md) (**Rev. PTKm**)

* We add more discussion about
   - Different ICL variants X-ICL in Appendix G (**Rev. PTKm**).
   - Performance saturation with longer ICL shots in Appendix A (**Rev dAEJ**)

---

### Meta-Review · Area_Chair_oepS · 2023-12-20

**Metareview:**

The paper presents an analysis of large multimodal models across five areas, examining the impact of in-context learning and proposing its new variants. Reviewers appreciated the comprehensive critique of contemporary models and the insights into improving them. There were concerns were raised about the lack of novel dataset or methodology contributions, and comprehensive analysis of ICL variants. However, I still recommend acceptance. The paper contributes to the community's understanding of model limitations and potential improvements.

**Justification For Why Not Higher Score:**

Lack of significant dataset or method contributions.

**Justification For Why Not Lower Score:**

Comprehensive critique of contemporary models.

---

### Decision · Program_Chairs · 2024-01-16

Accept (poster)